# CAKE: Cascading and Adaptive KV Cache Eviction with Layer Preferences

**Ziran Qin**[1,2*], **Yuchen Cao**[2], **Mingbao Lin**[3†], **Wen Hu**[2], **Shixuan Fan**[2], **Ke Cheng**[2],
**Weiyao Lin**[1†], **Jianguo Li**[2†]
[1]Shanghai Jiao Tong University, [2]Ant Group, [3]Independent Researcher

## Abstract

Large language models (LLMs) excel at processing long sequences, boosting demand for key-value (KV) caching. While recent efforts to evict KV cache have alleviated the inference burden, they often fail to allocate resources rationally across layers with different attention patterns. In this paper, we introduce **C**ascading and **A**daptive **KV** cache **E**viction (**CAKE**), a novel approach that frames KV cache eviction as a "cake-slicing problem." CAKE assesses layer-specific preferences by considering attention dynamics in both spatial and temporal dimensions, allocates rational cache size for layers accordingly, and manages memory constraints in a cascading manner. This approach enables a global view of cache allocation, adaptively distributing resources across diverse attention mechanisms while maintaining memory budgets. CAKE also employs a new eviction indicator that considers the shifting importance of tokens over time, addressing limitations in existing methods that overlook temporal dynamics. Comprehensive experiments on LongBench and NeedleBench show that CAKE maintains model performance with only 3.2% of the KV cache and consistently outperforms current baselines across various models and memory constraints, particularly in low-memory settings. Additionally, CAKE achieves over $10\times$ speedup in decoding latency compared to full cache when processing contexts of 128K tokens with FlashAttention-2. Our code is available at `https://github.com/antgroup/cakekv`.

## 1 Introduction

Large language models (LLMs) (Zhao et al., 2023; Achiam et al., 2023; Dubey et al., 2024; Anthropic, 2024; AI, 2024) have enhanced their long-text processing capabilities, improving performance in multi-turn dialogues (Chiang et al., 2023), document summarization (Zhang et al., 2024a), question answering (Kamalloo et al., 2023), and information retrieval (Liu et al., 2024c). New models such as GPT-4 (Achiam et al., 2023), Claude 3.5 (Anthropic, 2024), LLaMA 3.1 (Dubey et al., 2024) and Mistral Large 2 (AI, 2024) have extended token processing capacities beyond 128K. Expanded contexts necessitate a linear increase in key-value (KV) cache size, resulting in heavier inference-time memory burdens. Shazeer (2019); Ainslie et al. (2023) partially address this issue by merging key-value heads during the training phase. However, optimizing key-value cache without additional training is crucial for efficient inference of long contexts under memory constraints, particularly in typical deployment scenarios where the model structure is fixed.

One way to maintain a manageable KV cache size on the fly is to remove some KV pairs (Xiao et al., 2023; Zhang et al., 2024b; Li et al., 2024b). The idea is to eliminate less important KV pairs based on certain rules. Although recent methods have enhanced pair selection for removal, they typically assign uniform cache sizes across layers, disregarding layer-specific requirements. This approach can impair performance under memory constraints. Recent research is trying to improve this by looking at how attention works in different layers (Yang et al., 2024a; Cai et al., 2024). These methods, which depend on prior observations, might not work well for all input types and models. Another idea is to adjust the cache size based on the current attention pattern (Wan et al., 2024). This can help, but it may not always optimize overall performance.

---

*This work was done when Ziran Qin was a research intern at Ant Group.
†Corresponding Authors.

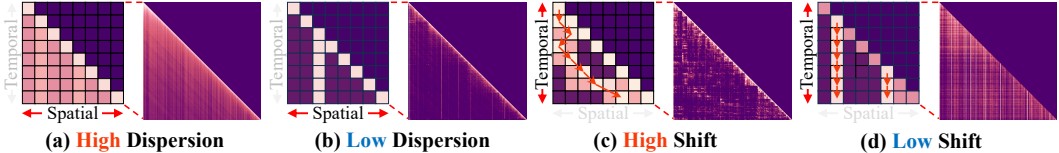

Figure 1: Variation in spatial (a, b) and temporal (c, d) characteristics of attention patterns. We provide toy examples (left) and real examples from Mistral's different layers (right) for illustration. For more detailed analysis and visualization of attention dynamics, please refer to Appendix J.

Allocating optimal cache sizes for different layers with a fixed memory budget is akin to "cake-slicing." The crux of this challenge is assessing each layer's affinity for KV cache, which is mirrored in its attention mechanisms. Our research delves into these mechanisms, examining spatial and temporal aspects across layers. Spatially, attention distribution across tokens can differ markedly between layers. Some layers may spread attention broadly, while others concentrate on specific tokens, as depicted in Figure 1(a) and Figure 1(b). Temporally, attention hotspots in certain layers shift over time in Figure 1(c), whereas in others, they remain constant in Figure 1(d). The variability of attention pattern necessitates a dynamic approach to memory allocation that goes beyond uniform (Zhang et al., 2024b; Liu et al., 2024d; Ren & Zhu, 2024; Li et al., 2024b), depicted in Figure 2(a), or fixed-pattern methods (Cai et al., 2024), drawn in Figure 2(b). They lack the flexibility to adapt to the nuanced attention dynamics. To enhance cache efficiency, a comprehensive strategy is required, factoring in each layer's attention pattern and KV cache preference. Moreover, conventional methods that evict KV pairs relying solely on static spatial attention scores often overlook the temporal evolution of attention, missing critical insights into how token relevance changes in various contexts.

In this paper, we introduce **C**ascading and **A**daptive **K**V cache **E**viction (CAKE), enhancing KV cache eviction by leveraging proposed preference-prioritized adaptive cache allocation strategy coupled with an innovative token eviction strategy. It is progressive and dynamic, employing a novel metric that assesses each layer's cache size preference, taking into account both the dispersion of spatial attention and the shifts in temporal attention, as outlined in Figure 2(c). To optimize cache allocation from a holistic viewpoint while keeping peak memory usage within desired limits, CAKE further employs a cascading memory management method. During the prefilling phase, CAKE dynamically manages the memory budgets for layers and adjusts the KV cache with the guidance of obtained preference scores, eliminating the need to store all KV cache for all layers. For token eviction, acknowledging the limitations of existing methods, we introduce an eviction indicator that considers sustained importance and attention variability, thereby minimizing the adverse effects of eviction on subsequent decoding steps. Extensive experiments have been conducted using various LLM architectures on the LongBench and NeedleBench benchmarks, encompassing nine major task categories. Results demonstrate CAKE's superior performance across diverse memory scenarios within these benchmarks. The allocation strategy in CAKE, which can complement and enhance existing KV cache eviction methods, offers a versatile solution for improved cache management. Furthermore, compared to full cache FlashAttention-2 implementation, CAKE significantly reduces memory consumption while simultaneously enhancing LLM throughput.

Our contributions are: (1) An analysis of attention dynamics revealing spatial dispersion and temporal shifts, leading to a layer-specific metric for cache size requirements. (2) An adaptive cache allocation strategy that optimizes overall cache allocation based on layer preferences. (3) A cascading cache management method that dynamically adjusts the KV cache during the prefilling stage, achieving memory usage efficiency comparable to uniform strategies without sacrificing eviction performance. (4) A new eviction indicator that considers the sustained importance and variability of tokens, enhancing token eviction performance.

## 2 BACKGROUND AND RELATED WORKS

**Basics of KV Cache Operations**. We revisit the fundamentals of KV caching. We focus on a single attention head, characterized as weight matrices $\mathbf{W}_Q, \mathbf{W}_K, \mathbf{W}_V \in \mathbb{R}^{D \times D}$, where $D$ denotes the model's hidden dimension. Considering a prompt embedding $\mathbf{X} \in \mathbb{R}^{S \times D}$, with $S$ representing the sequence length, an attention module has two phases: prompt prefilling and token decoding.

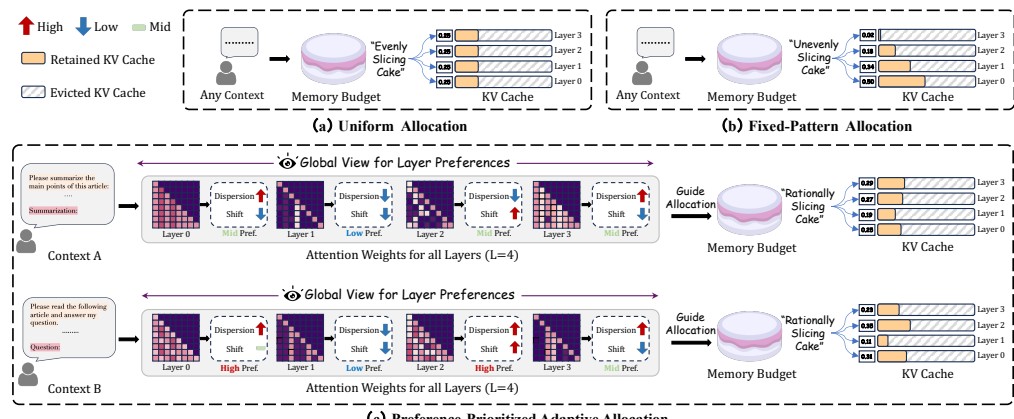

Figure 2: Illustration of CAKE compared with existing cache allocation strategies. (a) Uniform cache allocation (Xiao et al., 2023; Zhang et al., 2024b; Li et al., 2024b); (b) Fixed-shape cache allocation (Cai et al., 2024; Yang et al., 2024a); (c) Preference-prioritized adaptive cache allocation used in CAKE. Compared to (a) and (b), CAKE adjusts allocation ratios across different layers with layer preferences, adapting to various contexts and models, with given memory budgets.

*Prompt Prefilling*: The query, key, and value states are initially calculated as follows:
$$\mathbf{Q} = \mathbf{X}\mathbf{W}_Q, \quad \mathbf{K} = \mathbf{X}\mathbf{W}_K, \quad \mathbf{V} = \mathbf{X}\mathbf{W}_V. \tag{1}$$

Then, the output of the attention module is determined as:
$$\text{Attn}(\mathbf{Q}, \mathbf{K}, \mathbf{V}) = \mathbf{A}\mathbf{V}, \tag{2}$$

with attention weights $\mathbf{A} = \text{Softmax}(\frac{\mathbf{Q}\mathbf{K}^T}{\sqrt{D}}) \in \mathbb{R}^{S \times S}$. The key-value states $\mathbf{K}$ and $\mathbf{V}$ are then stored in a cache, establishing the KV cache.

*Token Decoding*: The KV cache is used and updated to produce new tokens. For each decoding step $i$, let the new token embedding be $\mathbf{x}_i \in \mathbb{R}^{1 \times D}$. To circumvent repeated key-value projections, the KV cache is concatenated with the newly generated KV pair $\mathbf{x}_i\mathbf{W}_K, \mathbf{x}_i\mathbf{W}_V \in \mathbb{R}^{1 \times D}$ for the current attention computation and to refresh the KV cache:
$$\mathbf{K} = \text{Concat}(\mathbf{K}, \mathbf{x}_i\mathbf{W}_K), \quad \mathbf{V} = \text{Concat}(\mathbf{V}, \mathbf{x}_i\mathbf{W}_V). \tag{3}$$

Though the KV cache alleviates the considerable computational demands of attention mechanisms, its linear growth with sequence length poses a challenge for extremely long input or output sequences. Efficiently managing the KV cache within a finite cache budget remains a formidable issue, especially in contexts that demand longer context lengths.

**KV Cache Eviction**. KV cache eviction during model inference enhances computational efficiency without altering the attention mechanism. This optimization relies on strategic cache reduction techniques. Early approaches to KV cache eviction focus on specific parts of the input sequence. For instance, StreamingLLM (Xiao et al., 2023) and LM-Infinite (Han et al., 2023) prioritize the retention of the first and last tokens. However, this strategy risks ignoring potentially important tokens in the middle of the sequence. Recognizing this limitation, subsequent research introduces more sophisticated indicators to filter unnecessary KV cache entries, such as using cumulative attention scores (Zhang et al., 2024b; Liu et al., 2024d), last token attention scores (Oren et al., 2024), mean attention scores (Ren & Zhu, 2024), or clustering recent attention scores (Li et al., 2024b). While these methods offer more nuanced approaches to cache eviction, they often apply the uniform allocation strategy, which may not maximize cache utilization. Recent FastGen (Ge et al., 2023) chooses which tokens to keep based on special markers, but it doesn't limit the total cache size. PyramidInfer (Yang et al., 2024a) and PyramidKV (Cai et al., 2024) allocate the budget in a pyramid-shaped manner, while D2O (Wan et al., 2024) adjusts cache size based on the current layer's attention density. While these methods have shown promise, they often rely on predefined cache allocation strategies across all layers or are just based on the local layer attention, which does not fully capture the complex dynamics of attention mechanisms and lacks a global perspective on layer preferences for the cache. In contrast, our work addresses these limitations by considering layer-specific cache preferences with a comprehensive and global perspective.

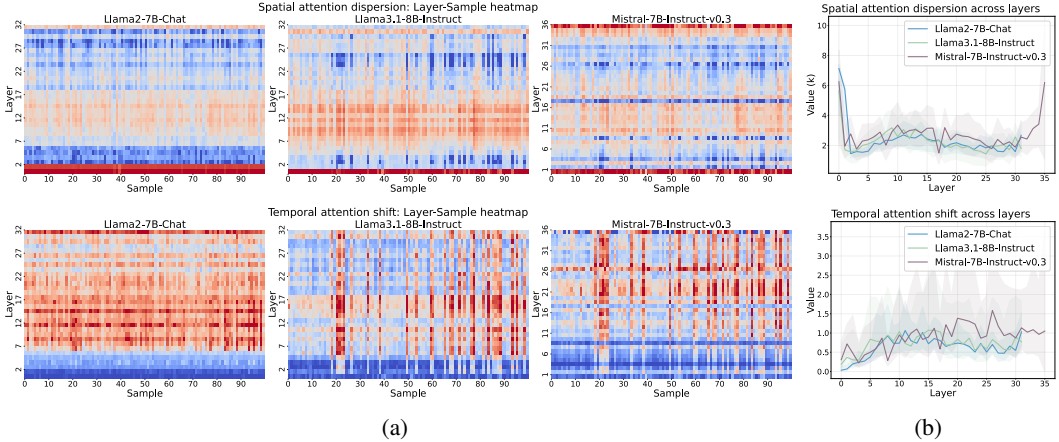

(a)                                          (b)

Figure 3: Analysis of attention dynamics. (a) Heatmap for spatial attention dispersion (upper) and temporal attention shift (lower), red color with high value, while blue for low value. The x-axis represents samples, and the y-axis represents layers. (b) Variation of spatial attention dispersion (upper) and temporal attention shift (lower) across layers and models. The experimental data is derived from the LongBench dataset (Bai et al., 2023).

## 3 INSIGHTS INTO ATTENTION DYNAMICS

Recent studies (Xiao et al., 2023; Zhang et al., 2024b) have shown that only a few KV cache elements are crucial during token decoding. Building on this, our analysis identifies two pivotal attention traits: *spatial attention dispersion*, showing how one token's attention spreads across others, and *temporal attention shift*, tracking the evolution of highly-attended tokens over time.

Recall some basics about the attention weights $\mathbf{A}$. Each $\mathbf{A}[i, j]$ shows how much the $i$-th query token attends to the $j$-th key token. We analyze the matrix spatially and temporally to explore attention dynamics in decoder-only models. (1) **Spatial Analysis**: Row $i$, $\mathbf{A}[i, :]$, shows how the $i$-th token's attention is distributed across tokens in one step. Examining $\mathbf{A}[i, :]$ reveals the attention landscape at that step. This helps reveal how the model prioritizes information at each generation step. (2) **Temporal Analysis**: Column $j$, $\mathbf{A}[:, j]$, shows how attention to the $j$-th token changes over steps. Each row in $\mathbf{A}[:, j]$ represents a time step, showing the $j$-th token's evolving attention. This dimension is key to tracking the model's shifting focus during sequence generation.

In this paper, we quantify the spatial attention dispersion by the entropy of $\mathbf{A}[i, :]$, defined as:

$$\mathrm{H}(\mathbf{A}) = -\sum_{i=0}^{S-1} \mathbf{A}[i, :] \log(\mathbf{A}[i, :])^T, \tag{4}$$

where the inner product is conducted between each row of $\mathbf{A}$ and its element-wise logarithm. Rows show higher values with even attention distribution and lower values with focused attention. Aggregating these values measures the evenness of attention distribution per token.

Accordingly, the temporal attention shift is characterized by the variance of $\mathbf{A}[:, j]$, expressed as:

$$\mathrm{V}(\mathbf{A}) = \sum_{j=0}^{S-1} \mathrm{Var}(\mathbf{A}[:, j]), \tag{5}$$

which computes the variance for each column of $\mathbf{A}$. Higher values indicate significant attention shifts across positions, while lower values suggest stable attention. Summing the variances column-wise captures the attention mechanism's temporal dynamics.

We analyze attention dynamics across multiple LLM layers using the LongBench dataset's long-context samples (Bai et al., 2023). Figure 3 (upper) shows attention dispersion changes across layers, and Figure 3 (lower) shows attention focus shifts. The results show significant variations in attention dispersion and shift across layers, models, and contexts. Despite recent KV cache advances (Xiao et al., 2023; Zhang et al., 2024b), our findings highlight LLM attention complexity, indicating a need for tailored KV cache management strategies to effectively handle the dynamic nature of attention.

## 4 METHODOLOGY

### 4.1 PREFERENCE-PRIORITIZED ADAPTIVE ALLOCATION

Given the variability in attention mechanisms across layers, models, and contexts, uniform or fixed-pattern cache allocation strategies prove inefficient, especially under limited memory budgets. To address these challenges, we introduce the preference-prioritized adaptive allocation strategy. It considers each layer's unique characteristics and adaptively allocates cache sizes from a global view of attention patterns. We define a preference metric for each layer's KV cache requirements, considering both the spatial dispersion and temporal shift of attention:

$$\mathcal{P} = \mathcal{H}^{\frac{1}{\tau_1}} \cdot \mathcal{V}^{\frac{1}{\tau_2}}, \quad \mathcal{H} = \mathrm{H}(\mathbf{A}[-S_w :, : -S_w]), \quad \mathcal{V} = \mathrm{V}(\mathbf{A}[-S_w :, : -S_w]), \tag{6}$$

where $\mathcal{P}$ is the preference score for an attention layer's cache size. As dispersed attention (high $\mathcal{H}$) requires retaining broader contexts, while frequent shifts (high $\mathcal{V}$) demand support for complex temporal patterns, layers with high preference score $\mathcal{P}$ benefit more from larger KV cache to maintain performance. Accounting for the varying importance of attention dispersion and shift across models and memory constraints, we introduce temperature parameters $\tau_1$ and $\tau_2$ to adjust their influence on $\mathcal{P}$. We focus on the submatrix $\mathbf{A}[-S_w :, : -S_w]$ of $\mathbf{A}$, representing a recent window of size $S_w$. This focus on the recent window chunk of prefilling attention weights is inspired by recent research (Li et al., 2024b; Yang et al., 2024a). These studies have demonstrated that decoding patterns can be effectively captured by analyzing the attention patterns of recent queries.

We implement the preference-prioritized adaptive allocation strategy using layer-specific preference scores. These scores adjust dynamically to varying models and contexts, ensuring adaptability. Each layer's cache size is set by normalizing the preference scores and allocating the total memory budget accordingly. This results in an adaptive cache size set $\mathbf{B} = \{B_l\}_{l=0}^{L-1}$, calculated as follows:

$$B_l = \frac{\mathcal{P}_l}{\sum_{k=0}^{L-1} \mathcal{P}_k} \cdot B_{\text{total}}, \tag{7}$$

where $\mathcal{P}_l$ is the preference score for layer $l$, and $B_{\text{total}}$ is the total cache budget. This method optimizes cache size allocation for each layer's unique characteristics and the input context.

### 4.2 PREFERENCE-GUIDED CASCADING CACHE MANAGEMENT

Our preference-prioritized adaptive allocation strategy, though optimal, initially requires a comprehensive view of prefilling attention weights across all layers. This leads to the peak memory usage of $\mathcal{O}(S \cdot L)$, which may exceed the cache budget $B_{\text{total}}$.

To overcome this, we further develop the preference-guided cascading cache management, as illustrated in Figure 4. It dynamically maintains the cache budget during prefilling, effectively reducing peak memory usage to the target. Algorithm 1 outlines our cache management procedure, which partitions the prefilling process into $L$ stages, one for each model layer, and cascadingly manages cache memory guided by preference scores. At each stage, as a new layer completes its prefilling computation, the budget is redistributed based on the current obtained preference scores (Algorithm 1, lines 3–7), and KV caches are updated among all processed layers according to the redistributed budget.

This approach ensures constant memory usage during the prefilling stage while maintaining equivalent cache distribution and eviction results as the standard strategy. Our algorithm is supported by the following theoretical results (Proposition 1 and Theorem 1, proofs provided in Appendix D).

**Proposition 1.** *For any layer $l \in [L]$, the allocated budget size decreases monotonically from stage $l$ to $L-1$:*

$$B_l^{(m+1)} < B_l^{(m)}, m \in [l, L-1], \tag{8}$$

*where $B_l^{(m)}$ is the redistributed cache budget for layer $l$ at stage $m$, calculated based on the current obtained preference score $\mathcal{P}$:*

$$B_l^{(m)} = \frac{\mathcal{P}_l}{\sum_{k=0}^{m} \mathcal{P}_k} \cdot B_{total}. \tag{9}$$

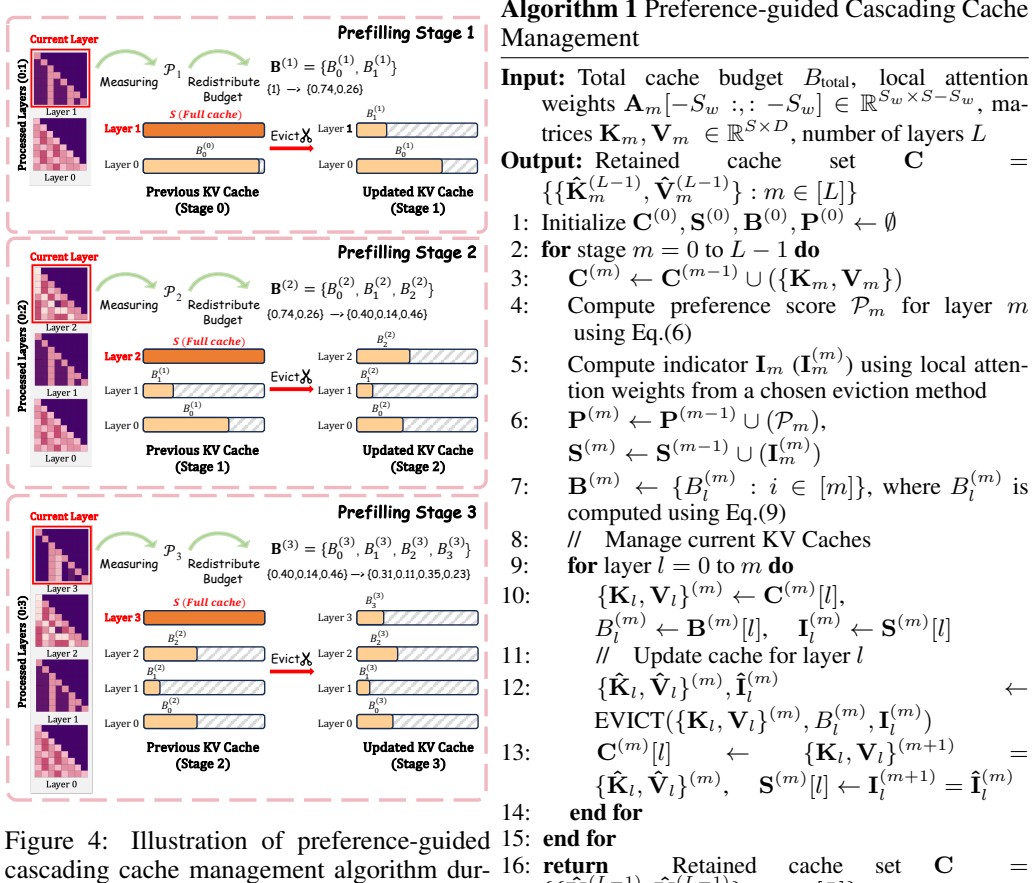

**Algorithm 1** Preference-guided Cascading Cache Management

**Input:** Total cache budget $B_{\text{total}}$, local attention weights $\mathbf{A}_m[-S_w :,: -S_w] \in \mathbb{R}^{S_w \times S - S_w}$, matrices $\mathbf{K}_m, \mathbf{V}_m \in \mathbb{R}^{S \times D}$, number of layers $L$

**Output:** Retained cache set $\mathbf{C} = \{\{\hat{\mathbf{K}}_m^{(L-1)}, \hat{\mathbf{V}}_m^{(L-1)}\} : m \in [L]\}$

1: Initialize $\mathbf{C}^{(0)}, \mathbf{S}^{(0)}, \mathbf{B}^{(0)}, \mathbf{P}^{(0)} \leftarrow \emptyset$
2: **for** stage $m = 0$ to $L - 1$ **do**
3:     $\mathbf{C}^{(m)} \leftarrow \mathbf{C}^{(m-1)} \cup (\{\mathbf{K}_m, \mathbf{V}_m\})$
4:     Compute preference score $\mathcal{P}_m$ for layer $m$ using Eq.(6)
5:     Compute indicator $\mathbf{I}_m$ ($\mathbf{I}_m^{(m)}$) using local attention weights from a chosen eviction method
6:     $\mathbf{P}^{(m)} \leftarrow \mathbf{P}^{(m-1)} \cup (\mathcal{P}_m)$,
    $\mathbf{S}^{(m)} \leftarrow \mathbf{S}^{(m-1)} \cup (\mathbf{I}_m^{(m)})$
7:     $\mathbf{B}^{(m)} \leftarrow \{B_l^{(m)} : i \in [m]\}$, where $B_l^{(m)}$ is computed using Eq.(9)
8:     //   Manage current KV Caches
9:     **for** layer $l = 0$ to $m$ **do**
10:       $\{\mathbf{K}_l, \mathbf{V}_l\}^{(m)} \leftarrow \mathbf{C}^{(m)}[l]$,
      $B_l^{(m)} \leftarrow \mathbf{B}^{(m)}[l]$,   $\mathbf{I}_l^{(m)} \leftarrow \mathbf{S}^{(m)}[l]$
11:       //   Update cache for layer $l$
12:       $\{\hat{\mathbf{K}}_l, \hat{\mathbf{V}}_l\}^{(m)}, \hat{\mathbf{I}}_l^{(m)} \leftarrow$
      EVICT($\{\mathbf{K}_l, \mathbf{V}_l\}^{(m)}, B_l^{(m)}, \mathbf{I}_l^{(m)}$)
13:       $\mathbf{C}^{(m)}[l] \leftarrow \{\mathbf{K}_l, \mathbf{V}_l\}^{(m+1)} = \{\hat{\mathbf{K}}_l, \hat{\mathbf{V}}_l\}^{(m)}$,   $\mathbf{S}^{(m)}[l] \leftarrow \mathbf{I}_l^{(m+1)} = \hat{\mathbf{I}}_l^{(m)}$
14:     **end for**
15: **end for**
16: **return** Retained cache set $\mathbf{C} = \{\{\hat{\mathbf{K}}_m^{(L-1)}, \hat{\mathbf{V}}_m^{(L-1)}\} : m \in [L]\}$

Figure 4: Illustration of preference-guided cascading cache management algorithm during three stages.

Proposition 1 guarantees that the cache budget allocated to each layer decreases monotonically as stages process, ensuring that the KV cache $\{\mathbf{K}_l, \mathbf{V}_l\}^{(m+1)}$ is always a proper subset of the previous stage's cache $\{\mathbf{K}_l, \mathbf{V}_l\}^{(m)}$, regardless of the eviction method used.

Before presenting Theorem 1, let's define some operational details related to our algorithm. Given indicator vector $\mathbf{I}_l \in \mathbb{R}^S$, it measures token importance for layer $l$. The specific calculation of our eviction indicator will be detailed in the subsequent section. For each stage $m \in [L]$, the eviction operation, denoted as EVICT($\cdot$), retains KV pairs $\hat{\mathbf{K}}_l^{(m)}, \hat{\mathbf{V}}_l^{(m)} \in \mathbb{R}^{B_l^{(m)} \times D}$ from the current cache $\mathbf{K}_l^{(m)}, \mathbf{V}_l^{(m)} \in \mathbb{R}^{B_l^{(m-1)} \times D}$ corresponding to the top-$B_l^{(m)}$ positions with the highest scores in $\mathbf{I}_l^{(m)} \in \mathbb{R}^{B_l^{(m-1)}}$. The indicator vector $\mathbf{I}_l^{(m)}$ is also updated to $\hat{\mathbf{I}}_l^{(m)} \in \mathbb{R}^{B_l^{(m)}}$, retaining only the elements corresponding to the preserved positions. The updated cache and indicator $\{\hat{\mathbf{K}}_l, \hat{\mathbf{V}}_l\}^{(m)}$, $\hat{\mathbf{I}}_l^{(m)}$ will serve as the basis $\{\mathbf{K}_l, \mathbf{V}_l\}^{(m+1)}$ and $\mathbf{I}_l^{(m+1)}$ for further updates in the next stage $m + 1$.

**Theorem 1.** *For any layer $l \in [L]$, the KV cache $\{\mathbf{K}_l, \mathbf{V}_l\}^{(L-1)}$ obtained through cascading eviction from stage $l$ to stage $L - 1$ is equivalent to the result of applying the eviction operation once on the full KV cache $\{\mathbf{K}_l, \mathbf{V}_l\} \in \mathbb{R}^{S \times D}$ with the cache budget $B_l$ computed in Eq.(7):*

$$\{\mathbf{K}_l, \mathbf{V}_l\}^{(L-1)} = \text{EVICT}(\{\mathbf{K}_l, \mathbf{V}_l\}^{(m)}, B_l^{(m)}, \mathbf{I}_l^{(m)})_{m=l}^{L-1} = \text{EVICT}(\{\mathbf{K}_l, \mathbf{V}_l\}, B_l, \mathbf{I}_l). \quad (10)$$

Theorem 1 demonstrates that our preference-guided cascading cache management achieves equivalent KV cache eviction results to the vanilla preference-prioritized adaptive allocation strategy, despite its cascading operation. This method is theoretically compatible with existing KV eviction techniques, a claim that will be empirically validated in Section 5.5.

### 4.3 ATTENTION-SHIFT TOLERANT EVICTION INDICATOR

Current top eviction indicators, such as accumulated or mean attention scores (Zhang et al., 2024b; Liu et al., 2024d; Ren & Zhu, 2024), simplify tokens' attention profiles into single values, effectively identifying important tokens. However, this approach may overlook tokens whose importance fluctuates, as it fails to capture the dynamics of shifting attention. This could result in prematurely evicting tokens from the cache, affecting the model's future ability to retrieve relevant information.

We propose a robust eviction strategy tolerant of attention shifts. We use a multi-faceted indicator considering sustained importance and attention variability. For layer $l$, the eviction indicator $\mathbf{I}_l \in \mathbb{R}^S$ is computed, where each element $\mathbf{I}_l[n]$ signifies the importance of token $n$, computed as:

$$\mathbf{I}_l[n] = \begin{cases} \mathrm{Mean}(\mathbf{A}_l[-S_w :, n]) + \gamma \cdot \mathrm{Var}(\mathbf{A}_l[-S_w :, n]), & \text{if} \quad n < S - S_w, \\ \Omega, & \text{otherwise,} \end{cases} \tag{11}$$

where $\mathrm{Mean}(\cdot)$ and $\mathrm{Var}(\cdot)$ measure sustained importance and attention variability, respectively, with $\gamma$ adjusting their influence. Inspired by recent research (Li et al., 2024b; Yang et al., 2024a), we assign an arbitrarily large value $\Omega$ to ensure the preservation of the most recent $W$ tokens. Also, a pooling layer clusters $\mathbf{I}_l[: S - S_w]$ to maintain context and prevent information fragmentation.

The eviction operation $\mathrm{EVICT}(\{\mathbf{K}_l, \mathbf{V}_l\}, B_l, \mathbf{I}_l)$ is then executed, retaining the KV pairs $\{\hat{\mathbf{K}}_l, \hat{\mathbf{V}}_l\}$ that correspond to the top-$B_l$ scores in $\mathbf{I}_l$:

$$\hat{\mathbf{K}}_l = \mathbf{K}_l[\mathbf{D}_l, :], \quad \hat{\mathbf{V}}_l = \mathbf{V}_l[\mathbf{D}_l, :], \quad \mathbf{D}_l = \mathrm{TopK}(\mathbf{I}_l, B_l), \tag{12}$$

where TopK selects the indices $\mathbf{D}_l \in \mathbb{R}^{B_l}$ of the top $B_l$ scores in $\mathbf{I}_l$.

## 5 EXPERIMENTATION

### 5.1 EXPERIMENTAL SETTINGS

**Backbone LLMs**. Our experiments involve five major open-source LLMs (7B-70B parameters) capable of handling 4k-128k token contexts. These models feature two representative attention structures: (1) *Multi-head attention*: including Llama2-Chat (Touvron et al., 2023) and Gemma-Instruct (Team et al., 2024). (2) *Grouped-query attention*: including Llama3-Instruct (Dubey et al., 2024), Mistral-v0.3 (Jiang et al., 2023), and Qwen2.5-Instruct (Team, 2024).

**Baseline Methods**. We assess CAKE's performance against five baselines: (1) *Uniform Allocation*: StreamingLLM (Xiao et al., 2023) balances initial and recent tokens, H2O (Zhang et al., 2024b) uses cumulative attention, TOVA (Oren et al., 2024) adopts last-token attention, and SnapKV (Li et al., 2024b) clusters recent attention. (2) *Non-Uniform Allocation*: PyramidKV (Cai et al., 2024) employs a fixed pyramid-shaped allocation strategy with SnapKV's eviction indicator.

**Evaluating Tasks**. To evaluate CAKE's performance across various memory budgets, we use two carefully designed benchmarks: (1) *LongBench* (Bai et al., 2023): Focuses on long-context understanding, encompassing 16 datasets in six categories: Single/Multi-Document QA, Summarization, Few-shot Learning, Synthetic Tasks, and Code Completion. (2) *NeedleBench* (Li et al., 2024a): Tests retrieval and reasoning in complex contexts through three subtasks: Single-Needle Retrieval, Multi-Needle Retrieval, and Multi-Needle Reasoning.

**Implementation Details**. CAKE is compared to baseline methods across different total memory budgets ranging from $64L$ to $2048L$. For uniform allocation, each layer has a fixed KV cache size. Non-uniform methods like PyramidKV and CAKE vary cache sizes per layer while keeping total memory constant. Tokens are evicted during both prefilling and decoding to maintain memory usage. Experiments are run on NVIDIA A100 80GB GPUs. For specifics, see the Appendix C.

### 5.2 EVALUATIONS ON LONGBENCH DATASET

We benchmark CAKE against other methods on 16 datasets. Figure 5 displays the performance of various methods under different memory constraints across three models. SnapKV surpasses legacy methods like StreamingLLM and H2O by using an observation window for memory retention. TOVA, relying solely on the last token, falls short by ignoring the broader context. Despite

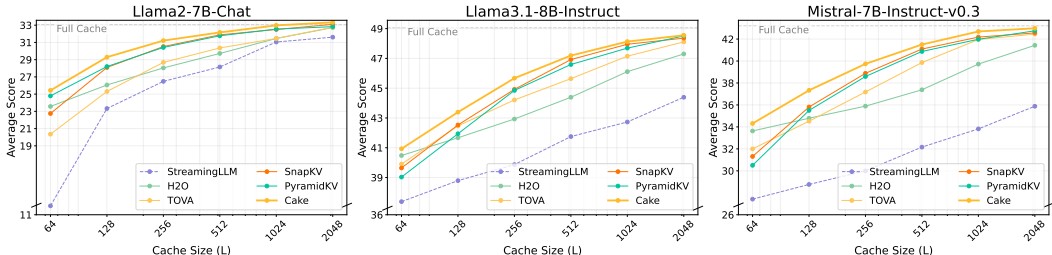

Figure 5: Average score among 16 datasets of LongBench under different cache budgets.

Table 1: Performance comparison over 16 datasets of LongBench. The best result is highlighted in **bold**, the second best in underline.

| Method | Single-Document QA | | | Multi-Document QA | | | Summarization | | | Few-shot Learning | | | Synthetic | | Code | | |
| | NrtvQA | Qasper | MF-en | HotpotQA | 2WikiMQA | Musique | GovReport | QMSum | MultiNews | TREC | TriviaQA | SAMSum | PCount | PR-en | Lcc | RB-P | Avg. |
|---|---|---|---|---|---|---|---|---|---|---|---|---|---|---|---|---|---|
| *Llama2-7B-Chat, $B_{total} = 128L$* | | | | | | | | | | | | | | | | | |
| StreamingLLM | 13.57 | 13.33 | 22.39 | 20.16 | 17.12 | 6.38 | 16.72 | 19.58 | 15.37 | 31.50 | 74.84 | 33.00 | 1.67 | 4.00 | 43.21 | 40.45 | 23.33 |
| H2O | 14.56 | 14.59 | 26.52 | 20.05 | 28.73 | 5.00 | 16.20 | 20.30 | 20.34 | 38.00 | 72.75 | 37.10 | 2.33 | 3.50 | 49.68 | 47.37 | 26.06 |
| TOVA | 13.56 | 15.46 | 20.28 | 22.22 | 25.01 | 4.79 | 15.72 | 19.11 | 15.58 | 42.50 | 79.58 | 37.06 | 3.45 | 4.61 | 46.34 | 39.70 | 25.31 |
| SnapKV | 13.70 | 16.27 | 32.52 | 22.76 | 28.59 | 7.56 | 18.87 | 19.96 | 20.05 | 43.50 | 79.90 | 36.74 | 2.79 | 7.00 | 51.86 | 47.41 | 28.09 |
| PyramidKV | 13.45 | 16.61 | 31.21 | 23.74 | 28.43 | 6.89 | 18.88 | 20.56 | 19.81 | 44.50 | 80.32 | 37.82 | 2.29 | 8.00 | 51.19 | 47.56 | 28.20 |
| CAKE | 15.15 | 16.38 | 32.17 | 25.25 | 31.09 | 7.00 | 19.47 | 21.07 | 20.36 | 48.50 | 80.66 | 37.77 | 4.42 | 9.00 | 51.48 | 48.90 | 29.29 |
| *Llama2-7B-Chat, $B_{total} = 1024L$* | | | | | | | | | | | | | | | | | |
| StreamingLLM | 13.73 | 16.75 | 26.28 | 25.97 | 25.67 | 5.87 | 22.21 | 19.27 | 24.07 | 61.00 | 80.88 | 40.90 | 2.28 | 1.00 | 56.71 | 48.79 | 29.46 |
| H2O | 16.58 | 17.67 | 32.04 | 26.01 | 28.73 | 7.81 | 22.87 | 20.99 | 25.05 | 60.00 | 83.02 | 40.06 | 2.90 | 3.50 | 57.57 | 52.02 | 31.05 |
| TOVA | 15.00 | 17.32 | 29.57 | 27.48 | 31.71 | 7.04 | 21.86 | 20.54 | 24.78 | 63.50 | 83.35 | 41.78 | 2.63 | 8.00 | 56.69 | 52.11 | 31.46 |
| SnapKV | 18.05 | 19.73 | 38.07 | 27.57 | 30.86 | 8.41 | 23.41 | 20.79 | 25.22 | 63.50 | 82.13 | 40.98 | 3.33 | 7.50 | 58.20 | 52.23 | 32.50 |
| PyramidKV | 17.94 | 20.68 | 38.85 | 27.25 | 29.99 | 7.91 | 23.17 | 21.55 | 25.45 | 63.50 | 82.97 | 40.79 | 3.57 | 7.50 | 57.72 | 51.93 | 32.55 |
| CAKE | 18.35 | 20.93 | 38.49 | 27.99 | 31.15 | 8.59 | 24.29 | 21.06 | 25.40 | 63.50 | 83.33 | 41.30 | 3.11 | 9.50 | 57.82 | 52.93 | 32.98 |
| *Llama3.1-8B-Instruct, $B_{total} = 128L$* | | | | | | | | | | | | | | | | | |
| StreamingLLM | 23.01 | 22.50 | 31.00 | 46.79 | 40.52 | 24.76 | 18.38 | 20.89 | 16.96 | 40.50 | 86.00 | 38.55 | 7.00 | 99.50 | 57.06 | 47.16 | 38.79 |
| H2O | 26.18 | 24.83 | 44.08 | 51.33 | 43.29 | 27.78 | 20.79 | 22.36 | 20.69 | 41.50 | 89.88 | 40.94 | 6.20 | 99.00 | 58.12 | 49.82 | 41.67 |
| TOVA | 29.44 | 27.03 | 45.20 | 54.15 | 44.87 | 28.87 | 22.21 | 20.54 | 20.52 | 53.50 | 92.27 | 40.99 | 6.17 | 99.50 | 51.14 | 43.15 | 42.47 |
| SnapKV | 26.40 | 27.67 | 47.83 | 53.40 | 44.20 | 28.68 | 20.56 | 23.11 | 15.58 | 45.50 | 89.36 | 39.98 | 6.33 | 99.00 | 57.69 | 50.69 | 42.53 |
| PyramidKV | 25.57 | 27.40 | 46.36 | 53.14 | 44.51 | 27.16 | 20.74 | 22.26 | 19.90 | 45.50 | 87.06 | 40.13 | 6.50 | 99.50 | 56.98 | 48.26 | 41.94 |
| CAKE | 27.41 | 32.01 | 49.58 | 52.99 | 45.76 | 28.32 | 21.85 | 23.15 | 21.56 | 47.50 | 89.61 | 40.90 | 6.33 | 99.50 | 57.92 | 49.83 | 43.39 |
| *Llama3.1-8B-Instruct, $B_{total} = 1024L$* | | | | | | | | | | | | | | | | | |
| StreamingLLM | 26.64 | 30.77 | 35.59 | 47.31 | 42.03 | 24.17 | 25.81 | 21.31 | 25.66 | 63.50 | 88.84 | 42.76 | 6.50 | 88.00 | 61.36 | 53.47 | 42.73 |
| H2O | 29.57 | 36.15 | 45.94 | 54.43 | 44.81 | 29.04 | 27.64 | 23.31 | 26.47 | 62.00 | 91.83 | 43.14 | 6.36 | 99.00 | 62.74 | 55.39 | 46.11 |
| TOVA | 30.66 | 40.95 | 51.09 | 54.58 | 46.51 | 30.62 | 28.12 | 23.61 | 26.24 | 68.00 | 91.49 | 43.80 | 5.92 | 99.50 | 60.73 | 52.64 | 47.15 |
| SnapKV | 30.95 | 44.74 | 52.58 | 55.09 | 46.83 | 30.37 | 27.87 | 24.57 | 25.99 | 68.00 | 92.03 | 42.60 | 6.50 | 99.50 | 63.00 | 56.50 | 47.95 |
| PyramidKV | 30.54 | 43.64 | 52.73 | 55.29 | 46.29 | 31.28 | 27.53 | 24.50 | 26.00 | 68.00 | 92.09 | 41.75 | 6.05 | 99.50 | 62.35 | 55.44 | 47.69 |
| CAKE | 30.88 | 44.95 | 52.38 | 55.49 | 46.99 | 30.82 | 28.68 | 24.91 | 26.39 | 69.00 | 91.94 | 42.60 | 6.00 | 99.50 | 62.65 | 56.89 | 48.13 |
| *Mistral-7B-Instruct-v0.3, $B_{total} = 128L$* | | | | | | | | | | | | | | | | | |
| StreamingLLM | 16.91 | 21.51 | 24.85 | 34.14 | 26.99 | 16.64 | 15.67 | 18.61 | 14.40 | 43.50 | 83.26 | 37.00 | 0.00 | 23.00 | 41.63 | 42.12 | 28.76 |
| H2O | 21.25 | 26.66 | 35.13 | 38.82 | 29.80 | 18.88 | 21.00 | 19.50 | 18.63 | 41.00 | 87.64 | 38.25 | 1.50 | 67.00 | 44.35 | 47.29 | 34.79 |
| TOVA | 22.47 | 24.26 | 37.22 | 42.26 | 28.85 | 19.97 | 19.40 | 18.70 | 17.86 | 63.00 | 88.98 | 37.71 | 0.50 | 56.50 | 37.42 | 37.22 | 34.52 |
| SnapKV | 21.02 | 27.26 | 41.25 | 45.15 | 29.23 | 22.75 | 20.47 | 20.17 | 17.75 | 42.50 | 87.28 | 38.01 | 0.50 | 69.50 | 44.48 | 45.69 | 35.81 |
| PyramidKV | 21.73 | 26.60 | 41.46 | 43.20 | 29.32 | 21.47 | 20.23 | 19.82 | 17.46 | 40.00 | 87.64 | 37.11 | 1.05 | 69.00 | 42.55 | 42.98 | 35.14 |
| CAKE | 22.31 | 29.15 | 43.51 | 44.51 | 30.36 | 22.85 | 21.56 | 20.47 | 18.96 | 47.00 | 88.60 | 39.36 | 1.00 | 76.50 | 44.96 | 46.19 | 37.33 |
| *Mistral-7B-Instruct-v0.3, $B_{total} = 1024L$* | | | | | | | | | | | | | | | | | |
| StreamingLLM | 20.96 | 28.05 | 30.03 | 37.06 | 27.56 | 16.03 | 24.03 | 19.07 | 22.79 | 67.00 | 87.61 | 40.96 | 1.50 | 21.50 | 48.05 | 48.87 | 33.82 |
| H2O | 23.78 | 31.63 | 41.31 | 43.24 | 31.07 | 20.43 | 26.74 | 20.41 | 23.93 | 67.50 | 88.84 | 42.62 | 1.50 | 72.00 | 50.60 | 49.87 | 39.72 |
| TOVA | 26.97 | 34.51 | 45.58 | 44.32 | 32.58 | 22.83 | 26.91 | 20.75 | 23.49 | 75.00 | 88.66 | 43.17 | 1.00 | 91.00 | 47.51 | 47.06 | 41.96 |
| SnapKV | 26.63 | 35.78 | 48.11 | 45.75 | 32.20 | 23.37 | 26.71 | 21.84 | 23.18 | 70.50 | 88.61 | 41.37 | 0.50 | 88.00 | 50.60 | 51.79 | 42.18 |
| PyramidKV | 25.51 | 36.02 | 47.72 | 44.74 | 33.16 | 23.91 | 26.55 | 21.83 | 23.27 | 70.50 | 88.41 | 40.94 | 1.00 | 87.00 | 50.17 | 50.93 | 41.98 |
| CAKE | 26.09 | 36.34 | 48.11 | 45.97 | 32.39 | 23.49 | 27.56 | 21.45 | 24.03 | 72.50 | 88.61 | 42.71 | 0.00 | 91.50 | 51.06 | 51.25 | 42.69 |

similarities to SnapKV, PyramidKV underperforms in many scenarios, possibly due to its limited adaptability. CAKE, however, outperforms others by dynamically allocating memory to each layer based on its specific needs, especially under tight memory conditions. Notably, CAKE achieves performance comparable to full cache models while only utilizing a small cache budget ($B_{total} > 512L$).

For detailed analysis, see Table 1 for results across two memory scenarios: a low setting ($B_{total} = 128L$) and a high setting ($B_{total} = 1024L$). Full results under different constraints are detailed in Appendix F.1. CAKE consistently ranks among the top performers or closely matches them across various tasks. In the low memory scenario with the Llama2 model, CAKE leads or finishes second in all datasets. CAKE's adaptive allocation strategy and robust eviction indicator consistently demonstrate superior and balanced performance, as its generalizability extends well to various model architectures with ranging model sizes. We evaluate this on two more architectures, Qwen2.5 and Gemma, as well as larger models (13B-70B), with results presented in Appendix F.2 and F.3.

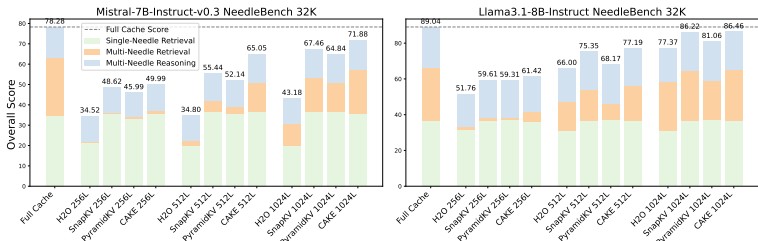

Figure 6: Performance comparison on NeedleBench 32K.

## 5.3 EVALUATIONS ON NEEDLEBENCH DATASET

We also test CAKE on retrieval and reasoning tasks using the NeedleBench dataset, with the results depicted in Figure 6. Consistent with the LongBench outcomes, CAKE shows superior overall performance. While utilizing only 3.2% of the cache size, CAKE preserves the model's capabilities for processing 32K-token-long contexts and even surpasses the full cache in the Single-Needle Retrieval task. This suggests that pruning specific cache information may eliminate superfluous details, allowing generation to focus on the most salient aspects and enhance its overall efficacy. Notably, CAKE outperforms existing methods in more complex tasks akin to real-world scenarios, particularly in Multi-Needle Retrieval. This can be attributed to CAKE's design, which balances long-term significance and short-term relevance, maintaining a more comprehensive and balanced representation of contextual information while avoiding premature discarding of information that could be vital for subsequent complex retrievals. For a more in-depth examination of the experimental outcomes, including detailed statistics and additional visualization graphs, please refer to Appendix G.

## 5.4 EVALUATION OF MEMORY AND THROUGHPUT

We assess the effectiveness of our method in reducing memory consumption and enhancing time efficiency during LLM inference by analyzing peak memory usage and decoding latency on Mistral-7B-Instruct-v0.3 implemented with FlashAttention-2 (Dao, 2023). Our comparison includes full cache, and three KV cache eviction methods, all with $B_{\text{total}} = 1024L$: SnapKV (uniform allocation strategy), PyramidKV (fixed-pattern allocation strategy), and our proposed CAKE method.

**Peak Memory Usage**. As depicted in the left panel of Figure 7, CAKE exhibits substantial memory-saving capabilities, comparable to other KV cache eviction methods such as uniform allocation (SnapKV) and fixed-pattern non-uniform allocation (PyramidKV). All these approaches maintain a fixed-size KV cache. When compared to the full cache implementation, CAKE achieves an impressive reduction in peak memory usage of approximate 48.63% with a 128K context length.

**Throughput Analysis**. Regarding decoding latency, the right panel of Figure 7 shows that both uniform and non-uniform eviction methods exhibit comparable inference performance. As input length increases, decoding latency for the full cache method grows significantly due to escalating computational demands and I/O latency bottlenecks. In contrast, CAKE maintains a relatively stable decoding speed by preserving a fixed amount of KV cache, resulting in significantly lower latency compared to the full cache, particularly for longer sequences. It is noteworthy that CAKE demonstrates remarkable efficiency, achieving over $10\times$ speedup in decoding latency compared to the full cache approach when processing sequences with 128K context length.

## 5.5 COMPATIBILITY WITH EXISTING KV EVICTION METHODS

Our preference-prioritized adaptive allocation strategy demonstrates strong compatibility with existing eviction indicators. We evaluate its performance using two representative methods: H2O, which serves as a classical indicator, and SnapKV, which represents a new advanced indicator. The experiments are conducted on LongBench datasets using Llama2-7B-Chat under $B_{\text{total}}$ of $128L$ and $512L$. We report the average performance across six tasks. In Figure 8, when compared with vanilla uniform cache allocation, methods equipped with our allocation strategy consistently improve performance across nearly all tasks. Importantly, they achieve significant overall performance gains. This comprehensive improvement across different eviction methods and tasks demonstrates the versatility and effectiveness of our allocation approach. It stresses the potential of our strategy as a generalizable framework for optimizing KV cache eviction, enhancing various existing methods.

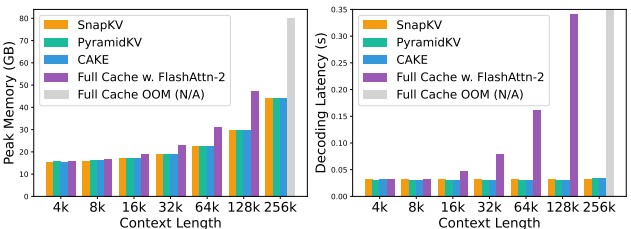

Figure 7: Peak memory usage and decoding latency on A100 80GB.

Table 2: Performance comparison with different allocation strategies. "P2A" denotes our preference-prioritized adaptive allocation.

| Strategy | Avg. |
|----------|------|
| Uniform | 28.36 |
| Pyramid | 28.69 |
| Random | 28.3±0.2 |
| P2A | **29.29** |

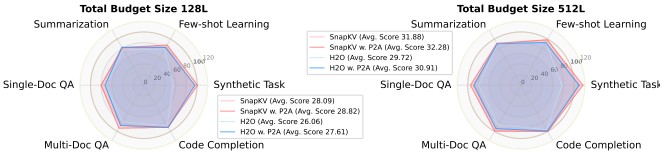

Figure 8: Performance on methods w/ or w/o preference-prioritized cache allocation strategy, abbreviated as "P2A."

Table 3: Performance comparison with different eviction indicators.

| Indicator | Avg. |
|-----------|------|
| Mean | 28.82 |
| Var | 28.68 |
| Mean · Var | 28.6 |
| Mean + Var | **29.29** |

## 5.6 ABLATION STUDIES

In this section, we present a series of ablation studies on LongBench to evaluate the effectiveness of our proposed allocation strategy and eviction indicator. We use Llama2-7B-Chat with cache budget $B_{\text{total}} = 128L$ as the default setting. Additional ablation studies are provided in Appendix I.

**Effectiveness of Proposed Allocation Strategy**. To assess our preference-prioritized adaptive allocation strategy, we compare it with various cache allocation strategies, including uniform, pyramid-shaped, and random allocation. As shown in Table 2, the pyramid-shape allocation strategy performs better than the uniform allocation strategy in this case. However, it may become ineffective in many scenarios, as we verified in the previous experimental section. Notably, the random method even outperforms the uniform strategy in some cases. This indicates that the conservative allocation strategy of uniform distribution fails to fully utilize the limited memory budget. Our allocation strategy consistently outperforms all baselines, as our method comprehensively measures the KV cache preference by capturing complex layer-specific attention patterns both spatially and temporally.

**Effectiveness of Proposed Eviction Indicator**. We further investigate effective eviction indicators to identify the most crucial tokens for preserving model performance. We compare different metrics for the eviction indicator: mean only, variance only, and combinations of mean and variance (multiplication and addition). The results in Table 3 suggest that, compared to variance only, tokens selected by mean attention are more conducive to maintaining performance. While the multiplicative combination of mean and variance slightly underperforms the individual metrics, the additive combination achieves the best performance overall. As we analyze, using mean attention effectively captures tokens with long-term importance, ensuring the retention of consistently relevant information. On the other hand, incorporating variance helps identify positions with the most significant changes, thus aiding in maintaining the attention distribution. The additive combination optimally balances these two aspects, leading to more informed eviction decisions.

## 6 CONCLUSION

In this paper, we propose Cascading and Adaptive KV cache Eviction (CAKE), a novel approach for optimizing KV cache evicting in LLMs. CAKE dynamically allocates cache sizes by leveraging a global view of layer-specific attention patterns, using cascading cache management guided by layer preferences. Additionally, CAKE introduces a new eviction indicator that accounts for both the long-term influence and temporal variability of token importance, allowing for more informed token selection. Experiments on the LongBench and NeedleBench benchmarks highlight CAKE's superior performance across different models and memory constraints, especially in low-memory scenarios. Our method enhances both LLM performance on long-context tasks and inference efficiency.

ACKNOWLEDGEMENT

The paper is supported in part by the National Natural Science Foundation of China (No.62325109, U21B2013) and Ant Group.

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

## A  ADDITIONAL RELATED WORKS

**Efficient Attention**. Numerous strategies have been devised to tackle the high time and space demands of the attention layer. These strategies include low-rank techniques to transform the original attention mechanism into one with linear or near-linear complexity (Choromanski et al., 2020; Wang et al., 2020; Katharopoulos et al., 2020; Alberti et al., 2023), sparse attention to confine the attention to specific, predetermined or adaptively learned patterns (Child et al., 2019; Kitaev et al., 2020; Roy et al., 2021), and their integration for a more comprehensive solution (Beltagy et al., 2020; Zaheer et al., 2020; Chen et al., 2021; Ren et al., 2021). Another group minimizes KV cache size. This is achieved by either re-engineering the architecture of multi-head attention or by innovating the update process of the KV cache. Multi-Query Attention (MQA) (Shazeer, 2019) and Group-Query Attention (GQA) (Ainslie et al., 2023) reduce KV heads by sharing KV pairs. Multi-head Latent Attention (MLA) (Liu et al., 2024a) furthers efficiency by compressing keys and values into latent vectors, while YOCO (Sun et al., 2024) employs cross-attention to recycle the shared KV cache. Dynamic Memory Compression (DMC) (Nawrot et al., 2024) dynamically consolidates the KV cache during the training process, thereby optimizing resource utilization.

**Orthogonal KV Cache Compression**. Beyond the eviction of the KV cache, numerous strategies have been investigated for its compression. Specifically, quantization methods (Liu et al., 2024e; Yue et al., 2024; Kang et al., 2024) have been applied to condense the KV cache into a lower-bitwidth format. Moreover, a fusion of KV cache eviction with quantization (Yang et al., 2024b), pruning Xu et al. (2024), low-rank factorization (Dong et al., 2024), or cache merging (Liu et al., 2024b; Wan et al., 2024) has been explored to achieve a more compact form of the cache, preserving some of the discarded data. Additionally, there are initiatives to transfer the evicted KV cache to auxiliary memory resources for potential redeployment in computation (Xiao et al., 2024; Fu et al., 2024). It is crucial to acknowledge that these compression techniques are compatible with KV cache eviction and, when integrated, can amplify their individual benefits, leading to more robust solutions for easing the KV cache load.

## B  LIMITATION AND FUTURE WORK

CAKE introduces a novel perspective on KV cache management by considering the model's varying attention patterns across layers and comprehensively measuring layer-specific preferences for KV cache allocation. This approach enables effective adaptation to different models and input contexts, optimizing the use of limited memory resources. Despite these strengths, several limitations and areas for future work remain. While CAKE improves cache management at the layer level, it does not address finer-grained dynamics within layers. Future work could explore integrating head-level attention patterns with layer-specific variations, potentially offering more nuanced control over cache allocation. Additionally, although we demonstrate CAKE's compatibility with existing eviction methods (Section 5.5) and quantization methods (Appendix 5.5), there is potential to further enhance memory efficiency by combining it with other KV cache optimization techniques such as cache merging (Liu et al., 2024b) and pruning (Xu et al., 2024). This integration could significantly expand CAKE's applicability and resource-saving capabilities. In the future, we will continue to explore strategies for efficient LLMs, focusing on both memory efficiency and computational optimization to further push the boundaries of resource-constrained inference.

## C  MORE IMPLEMENTATION DETAILS

In this section, we provide more detailed implementation information for CAKE. Our algorithm is divided into two phases: prompt prefilling and token decoding. During prompt prefilling, we utilize preference-guided cascading cache management (Section 4.2) based on the preference-prioritized adaptive allocation strategy (Section 4.1) to allocate appropriate cache sizes for different layers. During preference-guided cascading cache management, we manage the KV cache by updating it using eviction operation based on the attention-shift tolerant eviction indicator (Section 4.3). For the preference-prioritized adaptive allocation strategy, the temperature parameters $\tau_1$ and $\tau_2$ are set to $0.2 \sim 2$ and $0.4 \sim 3$ respectively, optimized through grid search. For our experimental models, we aggregate each quantized value to represent its layer characteristics. For the attention-shift tolerant eviction indicator, we fix $\gamma = 200$. Following SnapKV (Li et al., 2024b), we use an observation

window of size $S_w = 32$ and utilize a pooling layer clustering information. In the decoding stage, once the attention mechanisms for layers have been well established, we fix the budget size for each layer according to the allocation results obtained from the preference-prioritized adaptive allocation. Listing 1 provides PyTorch-style pseudo-code for CAKE's implementation with FlashAttention.

Listing 1: Implementation of CAKE in pseudo PyTorch style.

```python
class CakeCache(Cache):
    def __init__(self)
        self.pref_scores = []
        self.evict_indicators = []
        self.layers_budget = []
    ...
# Layer 0 to L-1
def attention_forward(self, hidden_states, ..., past_key_value:Optional[CakeCache] = None):
    ...
    # Compute query_states, key_states, value_states
    bsz, q_len, _ = hidden_states.size()
    ...
    # Compute local attention to the current layer
    local_attn = compute_local_attention(query_states[:,-self.window_size:,:], key_states,
     value_states)
    observed_attn = local_attn[:,:,-self.window_size:,:-self.window_size]
    if prefill:
        # Calculate preference score
        dispersion = calculate_attn_dispersion(observed_attn)
        shift = calculate_attn_shift(observed_attn)
        past_key_value.pref_scores[layer] = (dispersion**(1/tau1) * shift**(1/tau2))
        # Calculate eviction indicator
        attn_mean = mean(observed_attn)
        attn_var = var(observed_attn)
        indicator= attn_mean + self.gamma * attn_var
        indicator = pool1d(indicator)
        # Update
        past_key_value.pref_scores.append(pref_score)
        past_key_value.evict_indicators(pref_score, indicator)
        past_key_value = cake(past_key_value, q_len, self.layer_id)
    else:
        indicator= pool1d(observed_attn)
        past_key_value.update_score(indicator)
    ...
    # Compute attention output using flash_attention
    attn_output = flash_attention_forward(query_states, key_states, value_states, ...
    )
    return attn_output, past_key_value

def cake(self, past_key_value, seq_len, current_layer_idx):
    if seq_len<=self.cache_size-self.window_size:
        return past_key_values
    # Cache budget management
    if prefill:
        pref_scores = past_key_values.pref_scores
        layer_budgets = [pref_score/sum(pref_scores)*self.total_budget for pref_score in
     pref_scores]
        # The following loop can be executed in parallel for each layer
        for layer_idx, budget_size in enumerate(layer_budgets):
            budget_size = min(budget_size, seq_len - self.window_size)
            past_key_value = evict_layer_kvcache(past_key_value, layer_idx, budget_size)
            past_key_value.layers_budget[layer_idx] = budget_size
    else:
        budget_size = past_key_values.layer_budget[current_layer_idx]
        past_key_values = self.evcit_layer_kvcache(past_key_values, current_layer_idx,
     budget_size)
    return past_key_values

def evict_layer_kvcache(self, past_key_values, layer_idx, budget_size):
    bsz, num_key_value_heads, seq_len, head_dim = past_key_values.key_cache[layer_idx].shape
    num_key_value_groups = self.num_heads // num_key_value_heads
    indicator = past_key_values.evict_indicators[layer_idx]
    indices = indicator.topk(budget, dim=-1).indices
    # Update the eviction indicators
    past_key_values.evict_indicators[layer_idx] = compress_kv(indicator, indices, self.
     window_size)
    indices = indices.unsqueeze(-1).expand(-1, -1, -1, head_dim)
    # Update the KV cache
    past_key_values.key_cache[layer_idx] = compress_kv(key_states, indices, self.window_size)
    past_key_values.value_cache[layer_idx] = compress_kv(value_states, indices, self.
     window_size)
    return past_key_values
```

# D  PROOFS

*Proof of Proposition 1.*  Given the budget calculation formula in Eq.(9), we have:

$$B_l^{(m)} = \frac{\mathcal{P}_l}{\sum_{k=0}^{m} \mathcal{P}_k} \cdot B_{\text{total}}, \tag{13}$$

$$B_l^{(m+1)} = \frac{\mathcal{P}_l}{\sum_{k=0}^{m+1} \mathcal{P}_k} \cdot B_{\text{total}}. \tag{14}$$

And the inequality Eq.(8) is equivalent to:

$$\frac{\mathcal{P}_l}{\sum_{k=0}^{m+1} \mathcal{P}_k} < \frac{\mathcal{P}_l}{\sum_{k=0}^{m} \mathcal{P}_k}. \tag{15}$$

Since preference scores are always positive, this inequality holds. Therefore, we complete the proof that the allocated budget size monotonically decreases. $\square$

*Proof of Theorem 1.*  Recall that the eviction operation on layer $l$ essentially retains KV pairs based on top-$B_l$ selection indices $\mathbf{D}_l$ on the values of $\mathbf{I}_l$, where $B_l$ is the target budget size. Therefore, proving Eq.(10) is equivalent to demonstrating:

$$\text{TopK}(\mathbf{I}_l^{(m)}, B_l^{(m)})_{m=l}^{L-1} = \text{TopK}(\mathbf{I}_l, B_l). \tag{16}$$

The left-hand side of the above equation can be expanded as a cascading process:

$$\begin{aligned}
\mathbf{D}_l^{(l)} &= \text{TopK}(\mathbf{I}_l^{(l)}, B_l^{(l)}), \quad \mathbf{D}_l^{(l)} \in \mathbb{R}^{B_l^{(l)}}, \\
\mathbf{D}_l^{(l+1)} &= \text{TopK}(\mathbf{I}_l^{(l+1)}, B_l^{(l+1)}), \quad \mathbf{D}_l^{(l+1)} \in \mathbb{R}^{B_l^{(l+1)}}, \\
&\vdots \\
\mathbf{D}_l^{(L-1)} &= \text{TopK}(\mathbf{I}_l^{(L-1)}, B_l^{(L-1)}), \quad \mathbf{D}_l^{(L-1)} \in \mathbb{R}^{B_l^{(L-1)}}.
\end{aligned} \tag{17}$$

For the vanilla method, $\mathbf{D}_l$ is obtained by a single TopK operation:

$$\mathbf{D}_l = \text{TopK}(\mathbf{I}_l, B_l). \tag{18}$$

Given these definitions, our goal simplifies to proving that:

$$\mathbf{D}_l^{(L-1)} = \mathbf{D}_l. \tag{19}$$

To prove this equality, we first show that these two sets have the same number of elements. Note that $|\mathbf{D}_l^{(L-1)}| = B_l^{(L-1)}$ and $|\mathbf{D}_l| = B_l$. At the final stage, both methods have access to the preference scores of all layers, and the budget calculation for layer $l$ is identical in both cases:

$$B_l^{(L-1)} = B_l = \frac{\mathcal{P}_l}{\sum_{k=0}^{L-1} \mathcal{P}_k} \cdot B_{\text{total}}, \tag{20}$$

where $\mathcal{P}_l$ is the preference score for layer $l$, and $B_{\text{total}}$ is the total cache budget. Thus, $|\mathbf{D}_l^{(L-1)}| = |\mathbf{D}_l|$. Next, we further prove that elements in $\mathbf{D}_l^{(L-1)}$ and $\mathbf{D}_l$ are identical. Given that $B_l^{(m)}$ is decreasing as $m$ increases, for any stage $m \in [l, L-1)$, we have:

$$\text{TopK}(\mathbf{I}_l^{(m)}, B_l^{(m)}) \supset \text{TopK}(\mathbf{I}_l, B_l^{(m+1)}) = \text{TopK}(\mathbf{I}_l^{(m)}, B_l^{(m+1)}). \tag{21}$$

Therefore, we have:

$$\mathbf{D}_l^{(l)} \supset \mathbf{D}_l^{(l+1)} \supset ... \supset \mathbf{D}_l^{(L-1)} = \text{TopK}(\mathbf{I}_l^{(L-1)}, B_l^{(L-1)}) = \text{TopK}(\mathbf{I}_l, B_l). \tag{22}$$

In the final stage, set $\mathbf{D}_l^{(L-1)}$ contains exactly the $B_l^{(L-1)} = B_l$ highest-scoring elements from $\mathbf{I}_l$, which is identical to $\mathbf{D}_l$. Thus, we have proven that $\mathbf{D}_l^{(L-1)} = \mathbf{D}_l$, which completes the proof of Theorem 1.

$\square$

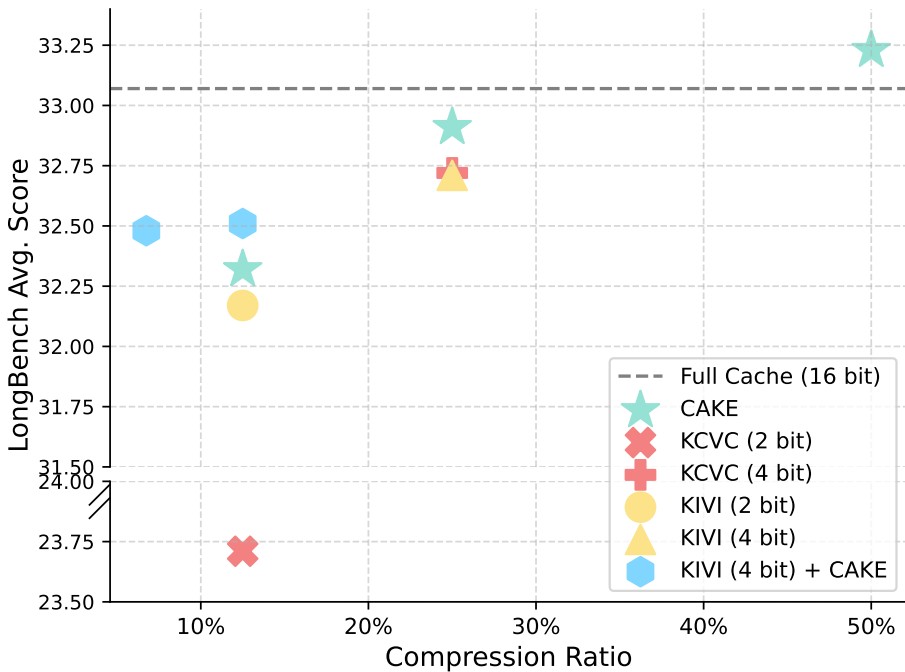

Figure 9: Comparison of CAKE, KV cache quantization methods (KIVI and KCVC), and their combination (KIVI with CAKE) on Llama2-7B-Chat using LongBench dataset.

## E    COMPARISON AND COMPATIBILITY WITH QUANTIZATION METHODS

While CAKE focuses on selectively dropping less important KV pairs to reduce memory footprint, quantization approaches aim to compress the cache by reducing the bit precision of stored values. These two strategies are orthogonal and potentially complementary in nature.

In this section, we evaluate CAKE against two quantization approaches: KIVI (Liu et al., 2024e), a state-of-the-art method using asymmetric KCVT quantization (quantizes Key cache per-channel and Value cache per-token), and a computationally efficient channel-wise quantization variant for both key and value caches (denoted as KCVC). As is shown in Figure 9, under matched compression ratios, CAKE outperforms full cache (33.23 *vs.* 33.07) and both quantization methods at 25% and 12.5% compression ratios. Notably, when combining KIVI-INT4 with CAKE, it demonstrates significant advantages in maintaining performance at high compression ratios. For instance, KIVI-INT4 with CAKE achieves 32.51 at 12.5% compression ratio, outperforming KIVI-INT2 (32.17) while using the same storage overhead. Even at a very aggressive 6.25% compression ratio, the combined approach still maintains strong performance (32.48), showing the effectiveness of integrating eviction and quantization strategies.

## F    EXTENDED EXPERIMENTAL RESULTS ON LONGBENCH

In this section, we provide comprehensive experimental results on LongBench (Bai et al., 2023), a benchmark focused on long-context understanding with input lengths ranging from 1,235 to 18,409 tokens. We present detailed performance evaluations across cache budgets from 64L to 2048L for three base models: Llama2-7B-Chat (Touvron et al., 2023), Llama3.1-8B-Instruct (Dubey et al., 2024), and Mistral-7B-Instruct-v0.3 (Jiang et al., 2023) (Appendix F.1). To demonstrate CAKE's generalizability, we conduct additional experiments on two more LLM architectures: Qwen2.5-7B-Instruct (Team, 2024) and Gemma-7B-Instruct (Team et al., 2024) (Appendix F.2). Furthermore, we evaluate CAKE's scalability on larger models ranging from 13B to 70B parameters, including Llama2-13B-Chat, Qwen2.5-32B-Instruct, and Llama3-70B-Instruct (Appendix F.3).

Table 4: LongBench Task Type.

| Task Type | #English Task | #Code Task |
|---|---|---|
| Multi-document QA | 3 | - |
| Single-document QA | 3 | - |
| Summarization | 3 | - |
| Few-shot learning | 3 | - |
| Synthetic Tasks | 2 | - |
| Code Completion | - | 2 |

LongBench is a long context dataset comprised of six task categories covering key long-text application scenarios: single-document QA, multi-document QA, summarization, few-shot learning, synthetic tasks and code completion. Our experiments are conducted on all of its 14 English subtasks and 2 code subtasks, spanning throughout all six categories (Table 4), with input lengths ranging from 1,235 to 18,409 tokens.

## F.1 DETAILED PERFORMANCE ANALYSIS ACROSS CACHE BUDGETS

Tables 5, 6 and 7 present the detailed LongBench results of CAKE and comparative methods applied to Llama2-7B-Chat, Llama3.1-8B-Instruct, and Mistral-7B-Instruct-v0.3, respectively.

**Results on Llama2-7B-Chat**. As shown in Table 5, CAKE consistently outperforms other KV eviction methods in terms of average score. Additionally, CAKE achieves top or second across each individual subtask and at all cache sizes. Notice that when using with $512L$ or more total budget size, CAKE achieves evaluation scores that are comparable with full cache across all subtasks.

**Results on Llama3.1-8B-Instruct**. The evaluation results are detailed in Table 6. The Llama3.1 series models have all incorporated the Grouped Query Attention (GQA) mechanism, and are able to support longer context length. The results show that CAKE continues to perform outstandingly across all subtasks in models based on the GQA structure.

**Results on Mistral-7B-Instruct-v0.3**. Table 7 contains the results of Mistral-7B-Instruct-v0.3. Around all 16 subtasks, CAKE shows consistent advantage throughout all the cache size budgets we have tested. Especially when the budget is limited, say $64L$, CAKE, equipped with the robust indicator and strategy, expresses relatively strong resistance to score decline.

The results as a whole demonstrate that CAKE, compared with other methods, consistently outperforms across all task types in LongBench when applied to the tested models and under various cache size budgets ranging from $64L$ to $2048L$. This demonstrates CAKE's effectiveness and broad applicability in KV cache-efficient long-context processing across various domains and across widely used open-source LLMs.

Table 5: Performance comparison over 16 datasets of LongBench on Llama2-7B-Chat for cache budgets from $64L$ to $2048L$. The best result is highlighted in **bold**, the second best in underline.

| Method | Single-Document QA | | | Multi-Document QA | | | Summarization | | | Few-shot Learning | | | Synthetic | | Code | | Avg. |
| --- | --- | --- | --- | --- | --- | --- | --- | --- | --- | --- | --- | --- | --- | --- | --- | --- | --- |
| | NrtvQA | Qasper | MF-en | HotpotQA | 2WikiMQA | Musique | GovReport | QMSum | MultiNews | TREC | TriviaQA | SAMSum | PCount | PR-en | Lcc | RB-P | |
| Llama2-7B-Chat, $B_{total} = Full$ | | | | | | | | | | | | | | | | | |
| Full | 20.68 | 20.19 | 36.69 | 27.83 | 31.45 | 8.21 | 27.09 | 20.71 | 26.24 | 64.0 | 83.09 | 41.41 | 2.94 | 7.75 | 58.51 | 52.29 | 33.07 |
| Llama2-7B-Chat, $B_{total} = 64L$ | | | | | | | | | | | | | | | | | |
| StreamingLLM | 7.35 | 16.62 | 6.84 | 10.73 | 14.68 | 2.84 | 13.34 | 18.42 | 12.76 | 17.50 | 19.38 | 11.44 | 0.25 | **4.50** | 20.87 | 14.91 | 12.03 |
| H2O | **12.77** | 13.76 | 21.00 | 19.16 | 26.53 | 4.47 | 14.09 | 20.01 | **18.49** | 32.00 | 70.92 | 34.79 | 2.75 | 3.00 | 42.09 | 41.36 | 23.57 |
| TOVA | 8.64 | 14.06 | 12.61 | 13.71 | 13.56 | 3.26 | 11.92 | 16.59 | 13.73 | **36.00** | 74.88 | 31.88 | 2.15 | 4.00 | 36.72 | 31.94 | 20.35 |
| SnapKV | 12.63 | 14.95 | 19.93 | 15.48 | 22.94 | 4.78 | 15.54 | 19.50 | 15.10 | 34.50 | 72.02 | 34.16 | 2.00 | 4.12 | 40.03 | 36.29 | 22.75 |
| PyramidKV | 10.94 | 15.53 | **28.43** | 23.53 | 28.82 | 5.10 | 15.33 | 19.79 | 15.29 | 34.50 | 76.16 | 34.84 | 2.88 | 2.56 | 44.82 | 38.16 | 24.79 |
| CAKE | 12.62 | **16.64** | 28.04 | 21.46 | 28.00 | **5.19** | 17.20 | 20.16 | 17.23 | 35.00 | 74.30 | **35.02** | 3.33 | **4.50** | 45.86 | 42.33 | 25.43 |
| Llama2-7B-Chat, $B_{total} = 128L$ | | | | | | | | | | | | | | | | | |
| StreamingLLM | 13.57 | 13.33 | 22.39 | 20.16 | 17.12 | 6.38 | 16.72 | 19.58 | 15.37 | 31.50 | 74.84 | 33.00 | 1.67 | 4.00 | 43.21 | 40.45 | 23.33 |
| H2O | 14.56 | 14.59 | 26.52 | 20.05 | 28.73 | 5.00 | 16.20 | 20.30 | 20.34 | 38.00 | 72.75 | 37.10 | 2.33 | 3.50 | 49.68 | 47.37 | 26.06 |
| TOVA | 13.56 | 15.46 | 20.28 | 22.22 | 25.01 | 4.79 | 15.72 | 19.11 | 15.58 | 42.50 | 79.58 | 37.06 | 3.45 | 4.61 | 46.34 | 39.70 | 25.31 |
| SnapKV | 13.70 | 16.27 | **32.52** | 22.76 | 28.59 | **7.56** | 18.87 | 19.96 | 20.05 | 43.50 | 79.90 | 36.74 | 2.79 | 7.00 | **51.86** | 47.41 | 28.09 |
| PyramidKV | 13.45 | **16.61** | 31.21 | 23.74 | 28.43 | 6.89 | 18.88 | 20.56 | 19.81 | 44.50 | 80.32 | **37.82** | 2.29 | 8.00 | 51.19 | 47.56 | 28.20 |
| CAKE | **15.15** | 16.38 | 32.17 | 25.25 | 31.09 | 7.00 | 19.47 | 21.07 | 20.36 | 48.50 | 80.66 | 37.77 | 4.42 | 9.00 | 51.48 | **48.90** | 29.29 |
| Llama2-7B-Chat, $B_{total} = 256L$ | | | | | | | | | | | | | | | | | |
| StreamingLLM | 13.20 | 13.45 | 24.36 | 21.50 | 24.08 | 5.74 | 17.28 | 18.97 | 18.12 | 44.00 | 78.43 | 37.55 | 1.58 | 4.50 | 53.85 | 47.03 | 26.48 |
| H2O | 15.82 | 14.72 | 27.26 | 21.07 | 27.00 | 6.32 | 19.75 | 20.29 | 22.23 | 45.50 | 79.64 | 37.93 | 2.93 | 4.00 | 53.91 | 50.33 | 28.04 |
| TOVA | 13.78 | 14.69 | 23.81 | 25.92 | 28.93 | 7.24 | 18.00 | 19.48 | 19.04 | 58.00 | 82.33 | **39.33** | 2.69 | 7.00 | 51.42 | 47.16 | 28.68 |
| SnapKV | 15.47 | 15.94 | 34.51 | 25.17 | 30.10 | 9.30 | 20.44 | 20.89 | 21.78 | 57.50 | 81.71 | 39.22 | 2.80 | 7.50 | 55.93 | 50.10 | 30.52 |
| PyramidKV | 15.13 | 15.86 | 33.93 | 25.58 | 29.51 | 9.23 | 20.35 | 21.22 | 22.01 | 58.00 | 82.39 | 38.47 | 2.15 | 7.50 | 55.81 | 49.53 | 30.42 |
| CAKE | **15.93** | **16.80** | **36.25** | **26.26** | 30.54 | 9.01 | **20.79** | 20.98 | 22.56 | **61.00** | 82.00 | 39.01 | 3.24 | 8.00 | **56.97** | 50.04 | 31.21 |
| Llama2-7B-Chat, $B_{total} = 512L$ | | | | | | | | | | | | | | | | | |
| StreamingLLM | 12.61 | 14.49 | 25.87 | 24.21 | 23.86 | 5.40 | 19.51 | 19.19 | 21.43 | 56.00 | 80.56 | 38.92 | 1.70 | 2.50 | 55.57 | 48.56 | 28.15 |
| H2O | 15.86 | 15.89 | 30.14 | 22.14 | 28.89 | 7.01 | 21.40 | 20.89 | 23.54 | 52.50 | 81.81 | 40.55 | 3.19 | 5.00 | 55.68 | 51.07 | 29.72 |
| TOVA | 13.63 | 15.47 | 26.26 | 26.55 | 31.25 | 7.63 | 19.81 | 19.88 | 22.19 | 62.50 | 83.10 | 40.63 | 2.72 | 8.00 | 55.83 | 50.28 | 30.36 |
| SnapKV | 16.42 | 17.19 | 36.52 | 26.32 | 31.15 | 9.09 | 21.33 | 21.11 | 23.58 | 64.00 | 81.82 | 39.29 | 3.33 | 8.50 | 58.00 | 52.46 | 31.88 |
| PyramidKV | 16.91 | 18.30 | 36.02 | 26.74 | 29.53 | 9.13 | 21.53 | 21.22 | 23.72 | 63.50 | 81.88 | 39.76 | 3.00 | 8.00 | 58.02 | 51.14 | 31.77 |
| CAKE | 16.36 | **19.31** | **37.82** | 26.51 | **31.34** | 8.10 | **21.77** | 21.42 | **24.36** | 64.00 | 82.63 | 39.62 | 3.33 | 8.00 | 57.73 | 52.33 | 32.16 |
| Llama2-7B-Chat, $B_{total} = 1024L$ | | | | | | | | | | | | | | | | | |
| StreamingLLM | 13.73 | 16.75 | 26.28 | 25.97 | 25.67 | 5.87 | 22.21 | 19.27 | 24.07 | 61.00 | 80.88 | 40.90 | 2.28 | 1.00 | 56.71 | 48.79 | 29.46 |
| H2O | 16.58 | 17.67 | 32.04 | 26.01 | 28.73 | 7.81 | 22.87 | 20.99 | 25.05 | 60.00 | 83.02 | 40.06 | 2.90 | 3.50 | 57.57 | 52.02 | 31.05 |
| TOVA | 15.00 | 17.32 | 29.57 | 27.48 | **31.71** | 7.04 | 21.86 | 20.54 | 24.78 | 63.50 | 83.35 | 41.78 | 2.63 | 8.00 | 56.69 | 52.11 | 31.46 |
| SnapKV | 18.05 | 19.73 | 38.07 | 27.57 | 30.86 | 8.41 | 23.41 | 20.79 | 25.22 | 63.50 | 82.13 | 40.98 | 3.33 | 7.50 | 58.20 | 52.23 | 32.50 |
| PyramidKV | 17.94 | 20.68 | **38.85** | 27.25 | 29.99 | 7.91 | 23.17 | 21.55 | 25.45 | 63.50 | 82.97 | 40.79 | 3.57 | 7.50 | 57.72 | 51.93 | 32.55 |
| CAKE | **18.35** | 20.93 | 38.49 | 27.99 | 31.15 | 8.59 | 24.29 | 21.06 | 25.40 | 63.50 | 83.33 | 41.30 | 3.11 | 9.50 | 57.82 | 52.93 | 32.98 |
| Llama2-7B-Chat, $B_{total} = 2048L$ | | | | | | | | | | | | | | | | | |
| StreamingLLM | 16.95 | 17.98 | 32.25 | 28.15 | 30.31 | 7.66 | 23.79 | 19.97 | 25.75 | 63.50 | 83.17 | 40.36 | 2.05 | 5.17 | 57.36 | 51.30 | 31.61 |
| H2O | 18.35 | 20.26 | 36.71 | 28.65 | 30.85 | 8.33 | 25.05 | 21.20 | 26.31 | 63.00 | 83.32 | 40.78 | 2.60 | 7.00 | 58.14 | 52.92 | 32.72 |
| TOVA | 18.73 | 20.52 | 35.38 | 28.10 | 30.91 | 8.87 | 24.86 | 20.87 | 26.22 | 64.00 | 83.13 | 41.99 | 2.75 | 7.00 | 58.07 | 52.77 | 32.76 |
| SnapKV | 18.80 | 22.18 | 38.18 | 27.72 | 30.92 | 8.37 | 25.87 | 21.35 | 25.92 | 64.00 | 83.13 | 40.95 | 3.32 | 8.50 | 58.18 | 52.23 | 33.10 |
| PyramidKV | 18.51 | 22.68 | 37.04 | 27.76 | 27.84 | 7.95 | **26.42** | 20.95 | 26.08 | 64.00 | 83.26 | 40.72 | 3.05 | 7.75 | **58.80** | 52.55 | 32.83 |
| CAKE | **19.25** | 22.16 | **38.70** | 27.72 | **31.58** | 8.91 | 26.19 | 20.70 | 26.19 | 64.00 | 83.25 | 41.02 | 3.86 | 9.00 | 57.92 | 52.76 | **33.33** |

Table 6: Performance comparison over 16 datasets of LongBench on Llama3.1-8B-Instruct for cache budgets from $64L$ to $2048L$ . The best result is highlighted in **bold**, the second best in underline.

| Method | Single-Document QA | | | Multi-Document QA | | | Summarization | | | Few-shot Learning | | | Synthetic | | Code | | Avg. |
|---|---|---|---|---|---|---|---|---|---|---|---|---|---|---|---|---|---|
| | NrtvQA | Qasper | MF-en | HotpotQA | 2WikiMQA | Musique | GovReport | QMSum | MultiNews | TREC | TriviaQA | SAMSum | PCount | PR-en | Lcc | RB-P | |
| *Llama3.1-8B-Instruct, $B_{total} = Full$* | | | | | | | | | | | | | | | | | |
| Full | 31.06 | 45.43 | 53.78 | 55.04 | 47.14 | 31.29 | 34.87 | 25.33 | 27.49 | 72.5 | 91.65 | 43.81 | 6.0 | 99.5 | 63.36 | 56.65 | 49.06 |
| *Llama3.1-8B-Instruct, $B_{total} = 64L$* | | | | | | | | | | | | | | | | | |
| StreamingLLM | 22.25 | 22.52 | 30.62 | 46.45 | 40.02 | 23.96 | 16.92 | 20.95 | 15.18 | 38.50 | 82.71 | 34.97 | 6.50 | 99.50 | 52.01 | 45.07 | 37.38 |
| H2O | 27.35 | 24.95 | 40.70 | 51.75 | 43.57 | 26.72 | 20.23 | 22.06 | 19.35 | 39.00 | 88.30 | 38.69 | 6.10 | 98.00 | 54.48 | 46.41 | 40.48 |
| TOVA | 27.14 | 23.83 | 41.82 | 53.54 | 43.07 | 27.42 | 20.06 | 16.67 | 18.11 | 47.50 | 89.82 | 39.12 | 6.50 | 98.00 | 47.41 | 38.46 | 39.90 |
| SnapKV | 26.09 | 24.06 | 41.22 | 51.32 | 45.11 | 25.71 | 17.38 | 21.47 | 15.81 | 38.50 | 84.98 | 37.36 | 6.50 | 99.50 | 53.16 | 46.09 | 39.64 |
| PyramidKV | 24.71 | 24.08 | 39.66 | 50.56 | 43.50 | 25.58 | 17.14 | 20.50 | 15.96 | 39.00 | 83.54 | 37.27 | 6.50 | 98.50 | 52.38 | 45.54 | 39.03 |
| CAKE | 27.03 | 26.56 | 42.96 | 52.42 | 45.00 | 27.73 | 21.29 | 18.57 | 18.26 | 40.50 | 88.42 | 39.31 | 6.07 | 99.00 | 54.38 | 47.35 | 40.93 |
| *Llama3.1-8B-Instruct, $B_{total} = 128L$* | | | | | | | | | | | | | | | | | |
| StreamingLLM | 23.01 | 22.50 | 31.00 | 46.79 | 40.52 | 24.76 | 18.38 | 20.89 | 16.96 | 40.50 | 86.00 | 38.55 | 7.00 | 99.50 | 57.06 | 47.16 | 38.79 |
| H2O | 26.18 | 24.83 | 44.08 | 51.33 | 43.29 | 27.78 | 20.79 | 22.36 | 20.69 | 41.50 | 89.88 | 40.94 | 6.20 | 99.00 | 58.12 | 49.82 | 41.67 |
| TOVA | 29.44 | 27.03 | 45.20 | 54.15 | 44.87 | 28.87 | 22.21 | 20.54 | 20.52 | 53.50 | 92.27 | 40.99 | 6.17 | 99.50 | 51.14 | 43.15 | 42.47 |
| SnapKV | 26.40 | 27.67 | 47.83 | 53.40 | 44.20 | 28.68 | 20.56 | 23.11 | 20.13 | 45.50 | 89.36 | 39.98 | 6.33 | 99.00 | 57.69 | 50.69 | 42.53 |
| PyramidKV | 25.57 | 27.40 | 46.36 | 53.14 | 44.51 | 27.16 | 20.74 | 22.26 | 19.90 | 45.50 | 87.06 | 40.13 | 6.50 | 99.50 | 56.98 | 48.26 | 41.94 |
| CAKE | 27.41 | 32.01 | 49.58 | 52.99 | 45.76 | 28.32 | 21.85 | 23.15 | 21.56 | 47.50 | 89.61 | 40.90 | 6.33 | 99.50 | 57.92 | 49.83 | 43.39 |
| *Llama3.1-8B-Instruct, $B_{total} = 256L$* | | | | | | | | | | | | | | | | | |
| StreamingLLM | 23.57 | 24.73 | 30.31 | 46.42 | 39.19 | 24.23 | 20.61 | 21.10 | 19.46 | 45.50 | 87.50 | 41.01 | 6.50 | 99.50 | 59.55 | 48.70 | 39.87 |
| H2O | 25.93 | 27.16 | 43.86 | 52.23 | 43.20 | 28.28 | 22.10 | 22.37 | 22.75 | 47.50 | 91.23 | 41.99 | 6.28 | 99.00 | 61.02 | 52.02 | 42.93 |
| TOVA | 29.70 | 30.85 | 48.80 | 54.36 | 46.71 | 30.00 | 23.21 | 22.18 | 22.88 | 57.50 | 91.83 | 42.77 | 6.05 | 99.50 | 54.83 | 46.15 | 44.21 |
| SnapKV | 27.22 | 35.25 | 50.96 | 54.50 | 45.73 | 28.56 | 22.75 | 23.80 | 22.83 | 55.50 | 91.28 | 41.42 | 6.03 | 99.50 | 60.64 | 52.80 | 44.92 |
| PyramidKV | 28.73 | 36.14 | 51.01 | 54.57 | 44.55 | 29.58 | 22.68 | 23.33 | 22.34 | 56.00 | 90.37 | 41.09 | 6.25 | 99.50 | 60.03 | 51.45 | 44.85 |
| CAKE | 29.75 | 38.74 | 52.27 | 54.52 | 45.39 | 29.57 | 24.15 | 24.24 | 23.77 | 57.00 | 90.97 | 41.82 | 6.10 | 99.50 | 61.01 | 51.90 | 45.67 |
| *Llama3.1-8B-Instruct, $B_{total} = 512L$* | | | | | | | | | | | | | | | | | |
| StreamingLLM | 25.64 | 27.48 | 33.30 | 47.36 | 40.06 | 24.80 | 23.16 | 20.80 | 22.85 | 57.50 | 87.69 | 42.08 | 6.50 | 97.00 | 60.51 | 51.28 | 41.75 |
| H2O | 27.76 | 29.01 | 44.75 | 52.78 | 44.31 | 29.22 | 24.71 | 23.11 | 24.56 | 54.50 | 91.38 | 42.10 | 6.36 | 99.00 | 62.30 | 54.33 | 44.39 |
| TOVA | 30.11 | 35.77 | 49.88 | 54.58 | 45.86 | 29.73 | 25.45 | 22.73 | 25.00 | 63.50 | 91.90 | 43.33 | 6.00 | 99.50 | 58.38 | 48.58 | 45.64 |
| SnapKV | 30.76 | 42.03 | 52.13 | 54.15 | 46.14 | 30.51 | 24.98 | 24.24 | 24.65 | 64.00 | 92.05 | 42.04 | 6.08 | 99.50 | 62.62 | 54.90 | 46.92 |
| PyramidKV | 30.47 | 42.15 | 52.17 | 54.67 | 45.25 | 30.60 | 25.00 | 24.33 | 24.51 | 62.50 | 91.24 | 41.67 | 5.95 | 99.50 | 61.58 | 53.89 | 46.59 |
| CAKE | 31.82 | 42.99 | 51.65 | 54.37 | 46.89 | 30.73 | 26.36 | 24.94 | 25.27 | 63.50 | 91.54 | 42.52 | 6.33 | 99.50 | 62.31 | 54.30 | 47.19 |
| *Llama3.1-8B-Instruct, $B_{total} = 1024L$* | | | | | | | | | | | | | | | | | |
| StreamingLLM | 26.64 | 30.77 | 35.59 | 47.31 | 42.03 | 24.17 | 25.81 | 21.31 | 25.66 | 63.50 | 88.84 | 42.76 | 6.50 | 88.00 | 61.36 | 53.47 | 42.73 |
| H2O | 29.57 | 36.15 | 45.94 | 54.43 | 44.81 | 29.04 | 27.64 | 23.31 | 26.47 | 62.00 | 91.83 | 43.14 | 6.36 | 99.00 | 62.74 | 55.39 | 46.11 |
| TOVA | 30.66 | 40.95 | 51.09 | 54.58 | 46.51 | 30.62 | 28.12 | 23.61 | 26.24 | 68.00 | 91.49 | 43.80 | 5.92 | 99.50 | 60.73 | 52.64 | 47.15 |
| SnapKV | 30.95 | 44.74 | 52.58 | 55.09 | 46.83 | 30.37 | 27.87 | 24.57 | 25.99 | 68.00 | 92.03 | 42.60 | 6.50 | 99.50 | 63.00 | 56.50 | 47.95 |
| PyramidKV | 30.54 | 43.64 | 52.73 | 55.29 | 46.29 | 31.28 | 27.53 | 24.50 | 26.00 | 68.00 | 92.09 | 41.75 | 6.05 | 99.50 | 62.35 | 55.44 | 47.69 |
| CAKE | 30.88 | 44.95 | 52.38 | 55.49 | 46.99 | 30.82 | 28.68 | 24.91 | 26.39 | 69.00 | 91.94 | 42.60 | 6.00 | 99.50 | 62.65 | 56.89 | 48.13 |
| *Llama3.1-8B-Instruct, $B_{total} = 2048L$* | | | | | | | | | | | | | | | | | |
| StreamingLLM | 27.40 | 36.91 | 37.85 | 49.23 | 44.66 | 24.31 | 28.57 | 21.67 | 27.12 | 67.50 | 90.98 | 42.49 | 6.12 | 87.00 | 63.06 | 55.39 | 44.39 |
| H2O | 29.65 | 39.53 | 48.64 | 54.23 | 46.50 | 29.28 | 29.97 | 23.68 | 27.21 | 68.00 | 91.48 | 43.06 | 6.11 | 99.50 | 63.06 | 56.91 | 47.30 |
| TOVA | 30.30 | 43.40 | 52.78 | 55.12 | 46.38 | 30.62 | 30.80 | 24.10 | 27.09 | 70.50 | 91.48 | 43.63 | 6.00 | 99.50 | 62.76 | 55.32 | 48.11 |
| SnapKV | 30.99 | 45.06 | 53.15 | 55.25 | 46.56 | 30.78 | 30.24 | 24.63 | 27.32 | 70.50 | 91.48 | 42.37 | 6.00 | 99.50 | 63.23 | 56.66 | 48.36 |
| PyramidKV | 31.13 | 45.06 | 53.80 | 55.78 | 46.59 | 30.89 | 30.25 | 24.82 | 27.35 | 71.00 | 91.65 | 42.62 | 6.00 | 99.50 | 63.27 | 56.44 | 48.51 |
| CAKE | 30.79 | 45.83 | 53.57 | 55.56 | 46.60 | 30.47 | 31.12 | 24.67 | 27.16 | 70.50 | 91.48 | 43.48 | 6.00 | 99.50 | 63.28 | 56.84 | 48.55 |

Table 7: Performance comparison over 16 datasets of LongBench on Mistral-7B-Instruct for cache budgets from $64L$ to $2048L$. The best result is highlighted in **bold**, the second best in underline.

| Method | Single-Document QA | | | Multi-Document QA | | | Summarization | | | Few-shot Learning | | | Synthetic | | Code | | Avg. |
|---|---|---|---|---|---|---|---|---|---|---|---|---|---|---|---|---|---|
| | NrtvQA | Qasper | MF-en | HotpotQA | 2WikiMQA | Musique | GovReport | QMSum | MultiNews | TREC | TriviaQA | SAMSum | PCount | PR-en | Lcc | RB-P | |
| *Mistral-7B-Instruct-v0.3, $B_{\text{total}} = Full$* | | | | | | | | | | | | | | | | | |
| Full | 28.26 | 39.33 | 49.47 | 45.29 | 33.21 | 24.57 | 32.25 | 22.16 | 24.58 | 75.5 | 88.36 | 43.78 | 0.5 | 83.0 | 49.79 | 51.3 | 43.21 |
| *Mistral-7B-Instruct-v0.3, $B_{\text{total}} = 64L$* | | | | | | | | | | | | | | | | | |
| StreamingLLM | 17.59 | 21.81 | 23.39 | 34.93 | 26.73 | 15.14 | 13.11 | 18.29 | 12.59 | 37.50 | 82.27 | 33.97 | 0.00 | 28.00 | 36.66 | 36.79 | 27.42 |
| H2O | 21.04 | 26.57 | 34.31 | 39.34 | 28.84 | 17.95 | 20.11 | 19.25 | 16.64 | 36.50 | 85.91 | 36.79 | 2.00 | 70.00 | 41.53 | 41.28 | 33.63 |
| TOVA | 21.87 | 23.48 | 34.36 | 41.60 | 28.35 | 19.46 | 17.79 | 18.41 | 15.24 | 50.50 | 89.16 | 37.18 | 0.00 | 48.50 | 32.78 | 33.28 | 32.00 |
| SnapKV | 20.63 | 24.03 | 34.69 | 41.22 | 29.94 | 18.04 | 15.42 | 18.82 | 12.62 | 36.50 | 86.44 | 36.70 | 1.00 | 46.50 | 37.77 | 40.65 | 31.31 |
| PyramidKV | 20.33 | 21.96 | 29.01 | 40.13 | 28.59 | 18.17 | 15.34 | 18.46 | 12.55 | 36.50 | 86.39 | 36.65 | 0.00 | 46.00 | 38.64 | 39.30 | 30.50 |
| CAKE | 21.83 | 26.85 | 35.19 | 39.70 | 28.89 | 19.70 | 20.12 | 19.19 | 17.21 | 38.00 | 86.49 | 37.02 | 0.50 | 75.00 | 41.17 | 42.18 | 34.31 |
| *Mistral-7B-Instruct-v0.3, $B_{\text{total}} = 128L$* | | | | | | | | | | | | | | | | | |
| StreamingLLM | 16.91 | 21.51 | 24.85 | 34.14 | 26.99 | 16.64 | 15.67 | 18.61 | 14.40 | 43.50 | 83.26 | 37.00 | 0.00 | 23.00 | 41.63 | 42.12 | 28.76 |
| H2O | 21.25 | 26.66 | 35.13 | 38.82 | 29.80 | 18.88 | 21.00 | 19.50 | 18.63 | 41.00 | 87.64 | 38.25 | 1.50 | 67.00 | 44.35 | 47.29 | 34.79 |
| TOVA | 22.47 | 24.26 | 37.22 | 42.26 | 28.85 | 19.97 | 19.40 | 18.70 | 17.86 | 63.00 | 88.98 | 37.71 | 0.50 | 56.50 | 37.42 | 37.22 | 34.52 |
| SnapKV | 21.02 | 27.26 | 41.25 | 45.15 | 29.23 | 22.75 | 20.47 | 20.17 | 17.75 | 42.50 | 87.28 | 38.01 | 0.50 | 69.50 | 44.48 | 45.69 | 35.81 |
| PyramidKV | 21.73 | 26.60 | 41.46 | 43.20 | 29.32 | 21.47 | 20.23 | 19.82 | 17.46 | 40.00 | 87.64 | 37.11 | 1.05 | 69.00 | 42.55 | 42.98 | 35.14 |
| CAKE | 22.31 | 29.15 | 43.51 | 44.51 | 30.36 | 22.85 | 21.56 | 20.47 | 18.96 | 47.00 | 88.60 | 39.36 | 1.00 | 76.50 | 44.96 | 46.19 | 37.33 |
| *Mistral-7B-Instruct-v0.3, $B_{\text{total}} = 256L$* | | | | | | | | | | | | | | | | | |
| StreamingLLM | 18.10 | 23.41 | 25.99 | 34.55 | 26.78 | 16.25 | 17.97 | 18.82 | 16.56 | 52.50 | 84.26 | 39.42 | 1.00 | 16.50 | 44.76 | 44.99 | 30.12 |
| H2O | 22.36 | 27.14 | 37.79 | 39.21 | 30.25 | 19.79 | 22.35 | 19.40 | 20.06 | 50.50 | 87.34 | 39.49 | 1.50 | 62.00 | 47.01 | 48.11 | 35.89 |
| TOVA | 22.52 | 27.29 | 41.32 | 44.49 | 31.08 | 22.36 | 21.99 | 18.96 | 20.00 | 68.00 | 88.58 | 39.76 | 0.50 | 66.00 | 40.71 | 41.35 | 37.18 |
| SnapKV | 23.06 | 29.43 | 44.61 | 45.05 | 30.39 | 22.89 | 22.32 | 20.77 | 20.19 | 56.00 | 88.21 | 39.34 | 0.00 | 83.00 | 48.09 | 48.72 | 38.88 |
| PyramidKV | 22.66 | 28.85 | 45.84 | 44.30 | 30.43 | 21.78 | 22.48 | 20.97 | 19.94 | 53.00 | 88.41 | 38.88 | 1.05 | 85.00 | 46.11 | 47.78 | 38.59 |
| CAKE | 24.01 | 30.78 | 45.88 | 45.06 | 31.45 | 23.43 | 22.81 | 20.87 | 20.92 | 58.50 | 88.29 | 40.16 | 0.50 | 87.00 | 47.76 | 48.43 | 39.74 |
| *Mistral-7B-Instruct-v0.3, $B_{\text{total}} = 512L$* | | | | | | | | | | | | | | | | | |
| StreamingLLM | 19.93 | 25.83 | 28.25 | 36.21 | 27.60 | 15.41 | 21.23 | 18.67 | 19.50 | 64.00 | 86.45 | 39.75 | 1.00 | 16.50 | 46.70 | 47.57 | 32.16 |
| H2O | 22.34 | 28.01 | 38.32 | 40.12 | 30.25 | 18.38 | 24.37 | 20.38 | 21.80 | 59.50 | 87.33 | 41.30 | 2.00 | 65.50 | 48.68 | 49.76 | 37.38 |
| TOVA | 25.06 | 31.69 | 44.22 | 44.90 | 32.48 | 23.44 | 24.33 | 19.81 | 21.77 | 71.50 | 88.46 | 42.39 | 0.00 | 77.50 | 44.54 | 45.66 | 39.86 |
| SnapKV | 24.88 | 32.92 | 47.31 | 45.23 | 31.78 | 23.08 | 24.60 | 21.80 | 21.52 | 65.50 | 88.61 | 40.43 | 0.50 | 88.50 | 50.59 | 50.22 | 41.09 |
| PyramidKV | 24.15 | 33.10 | 46.62 | 44.16 | 32.14 | 23.37 | 24.22 | 21.15 | 21.29 | 68.00 | 88.49 | 40.27 | 1.55 | 87.50 | 48.74 | 49.17 | 40.87 |
| CAKE | 25.82 | 34.24 | 46.93 | 45.34 | 31.71 | 23.38 | 25.81 | 21.47 | 22.41 | 68.00 | 88.46 | 41.27 | 0.50 | 90.00 | 49.04 | 49.67 | 41.50 |
| *Mistral-7B-Instruct-v0.3, $B_{\text{total}} = 1024L$* | | | | | | | | | | | | | | | | | |
| StreamingLLM | 20.96 | 28.05 | 30.03 | 37.06 | 27.56 | 16.03 | 24.03 | 19.07 | 22.79 | 67.00 | 87.61 | 40.96 | 1.50 | 21.50 | 48.05 | 48.87 | 33.82 |
| H2O | 23.78 | 31.63 | 41.31 | 43.24 | 31.07 | 20.43 | 26.74 | 20.41 | 23.93 | 67.50 | 88.84 | 42.62 | 1.50 | 72.00 | 50.60 | 49.87 | 39.72 |
| TOVA | 26.97 | 34.51 | 45.58 | 44.32 | 32.58 | 22.83 | 26.91 | 20.75 | 23.49 | 75.00 | 88.66 | 43.17 | 1.00 | 91.00 | 47.51 | 47.06 | 41.96 |
| SnapKV | 26.63 | 35.78 | 48.11 | 45.75 | 32.20 | 23.37 | 26.71 | 21.84 | 23.18 | 70.50 | 88.61 | 41.37 | 0.50 | 88.00 | 50.60 | 51.79 | 42.18 |
| PyramidKV | 25.51 | 36.02 | 47.72 | 44.74 | 33.16 | 23.91 | 26.55 | 21.83 | 23.27 | 70.50 | 88.41 | 40.94 | 1.00 | 87.00 | 50.17 | 50.93 | 41.98 |
| CAKE | 26.09 | 36.34 | 48.11 | 45.97 | 32.39 | 23.49 | 27.56 | 21.45 | 24.03 | 72.50 | 88.61 | 42.71 | 0.00 | 91.50 | 51.06 | 51.25 | 42.69 |
| *Mistral-7B-Instruct-v0.3, $B_{\text{total}} = 2048L$* | | | | | | | | | | | | | | | | | |
| StreamingLLM | 22.60 | 32.48 | 33.26 | 40.53 | 29.57 | 15.69 | 26.64 | 19.70 | 23.64 | 69.00 | 88.41 | 42.44 | 2.00 | 28.50 | 48.69 | 50.85 | 35.88 |
| H2O | 25.76 | 34.28 | 44.42 | 44.41 | 32.17 | 20.14 | 28.79 | 21.23 | 24.38 | 71.00 | 88.41 | 43.49 | 1.50 | 82.00 | 50.51 | 50.39 | 41.43 |
| TOVA | 27.48 | 37.20 | 47.47 | 45.25 | 33.14 | 23.73 | 29.37 | 21.73 | 24.52 | 75.00 | 88.66 | 43.33 | 1.00 | 82.00 | 49.08 | 50.18 | 42.45 |
| SnapKV | 26.43 | 37.60 | 48.27 | 45.37 | 33.08 | 24.02 | 28.81 | 21.43 | 24.40 | 73.50 | 88.36 | 42.34 | 1.00 | 85.50 | 49.62 | 51.28 | 42.56 |
| PyramidKV | 26.00 | 37.63 | 48.25 | 45.82 | 37.13 | 24.00 | 28.90 | 22.13 | 24.35 | 72.50 | 88.36 | 42.28 | 1.00 | 84.50 | 50.06 | 51.07 | 42.75 |
| CAKE | 25.43 | 37.84 | 48.30 | 45.60 | 32.21 | 24.14 | 30.00 | 21.83 | 24.35 | 74.50 | 88.36 | 43.38 | 0.50 | 89.00 | 50.68 | 51.48 | 42.98 |

Table 8: Performance comparison over 16 datasets of LongBench on Qwen2.5-7B-Instruct and Gemma-7B-Instruct. The best result is highlighted in **bold**, the second best in underline.

| Method | Single-Document QA | | | Multi-Document QA | | | Summarization | | | Few-shot Learning | | | Synthetic | | Code | | Avg. |
|---|---|---|---|---|---|---|---|---|---|---|---|---|---|---|---|---|---|
| | NrtvQA | Qasper | MF-en | HotpotQA | 2WikiMQA | Musique | GovReport | QMSum | MultiNews | TREC | TriviaQA | SAMSum | PCount | PR-en | Lcc | RB-P | |
| Qwen2.5-7B-Instruct, $B_{\text{total}} = Full$ | | | | | | | | | | | | | | | | | |
| Full | 29.05 | 43.34 | 52.52 | 57.59 | 47.05 | 30.24 | 31.78 | 23.64 | 23.96 | 72.5 | 89.47 | 45.61 | 8.5 | 100.0 | 59.61 | 67.12 | 48.87 |
| Qwen2.5-7B-Instruct, $B_{\text{total}} = 128L$ | | | | | | | | | | | | | | | | | |
| StreamingLLM | 18.80 | 23.57 | 26.71 | 40.69 | 37.20 | 16.24 | 15.70 | 18.01 | 12.61 | 42.50 | 81.22 | 40.10 | 8.50 | 25.00 | 46.77 | 47.47 | 31.32 |
| H2O | 22.73 | 24.52 | 31.04 | 43.89 | 40.19 | 18.92 | 18.28 | 19.82 | **16.03** | 42.00 | 83.11 | 41.88 | 8.50 | 96.50 | 50.39 | 55.05 | 38.30 |
| TOVA | 21.01 | 23.98 | 33.65 | 47.65 | 38.97 | 19.42 | **19.84** | 19.41 | 15.54 | 51.50 | 85.43 | 42.49 | 8.50 | 93.50 | 42.60 | 44.48 | 38.00 |
| SnapKV | 23.85 | 27.19 | 42.43 | 51.31 | 42.59 | 24.38 | 18.19 | 19.76 | 14.54 | 42.50 | 84.24 | 41.10 | 9.00 | 97.00 | 49.45 | 53.95 | 40.09 |
| PyramidKV | 19.56 | 23.39 | 37.72 | 48.33 | 38.33 | 19.53 | 15.10 | 19.32 | 12.05 | 42.00 | 83.29 | 38.84 | 9.00 | 84.50 | 47.87 | 50.39 | 36.83 |
| CAKE | **25.99** | **33.00** | **46.22** | **54.57** | **45.55** | **25.59** | 19.51 | **20.95** | 15.82 | **43.00** | 84.88 | 40.99 | 8.50 | 96.50 | **50.62** | **55.27** | **41.68** |
| Qwen2.5-7B-Instruct, $B_{\text{total}} = 1024L$ | | | | | | | | | | | | | | | | | |
| StreamingLLM | 22.72 | 29.42 | 31.47 | 43.57 | 38.18 | 17.99 | 24.33 | 19.47 | 22.46 | 61.00 | 87.53 | 43.79 | **8.50** | 34.00 | 55.17 | 58.43 | 37.38 |
| H2O | 26.45 | 34.94 | 40.49 | 48.63 | 42.02 | 22.27 | 25.67 | 20.90 | 22.41 | 59.00 | 87.83 | 45.07 | **8.50** | 98.48 | **59.77** | 63.88 | 44.14 |
| TOVA | 26.36 | 37.97 | 47.88 | 55.32 | 45.72 | 28.38 | 26.15 | 21.64 | 22.70 | **69.50** | 88.64 | 44.23 | **8.50** | 100.00 | 56.66 | 60.07 | 46.23 |
| SnapKV | 29.24 | 41.61 | 50.93 | **57.60** | 45.50 | 29.39 | 25.63 | 23.06 | 22.26 | 65.50 | 88.92 | 44.65 | **8.50** | 100.00 | 58.16 | **65.30** | 47.27 |
| PyramidKV | 29.34 | 38.60 | 50.17 | 55.67 | 45.12 | 27.82 | 23.26 | 22.16 | 20.55 | 62.50 | 86.85 | 43.26 | **8.50** | 100.00 | 57.76 | 61.99 | 45.85 |
| CAKE | **29.47** | **42.71** | **52.12** | 56.11 | **46.41** | **29.13** | **26.86** | **23.12** | **22.72** | 67.50 | **89.23** | 45.46 | **8.50** | 100.00 | 59.11 | 64.79 | **47.70** |
| Gemma-7B-Instruct, $B_{\text{total}} = Full$ | | | | | | | | | | | | | | | | | |
| Full | 15.33 | 34.27 | 47.43 | 28.2 | 22.69 | 7.48 | 26.64 | 19.67 | 23.84 | 69.0 | 79.41 | 32.59 | 1.0 | 44.5 | 47.39 | 45.96 | 34.09 |
| Gemma-7B-Instruct, $B_{\text{total}} = 128L$ | | | | | | | | | | | | | | | | | |
| StreamingLLM | 10.65 | 20.76 | 27.41 | 24.74 | 20.06 | 5.91 | 13.53 | 16.98 | 14.30 | 47.00 | 74.51 | 29.16 | **4.00** | 9.50 | 46.68 | 45.70 | 25.68 |
| H2O | 12.98 | 27.27 | 35.97 | 27.70 | 23.58 | 7.21 | 17.70 | 18.17 | 18.23 | 44.00 | 79.12 | 31.84 | 1.63 | 46.50 | 47.28 | 48.62 | 30.49 |
| TOVA | 13.73 | 27.38 | 42.38 | 28.73 | 23.34 | 7.80 | 17.06 | 17.76 | 17.61 | 57.50 | 79.65 | 31.35 | 1.50 | 49.00 | 41.85 | 43.22 | 31.24 |
| SnapKV | 12.98 | 27.53 | 42.90 | 29.69 | 25.45 | 7.01 | 17.14 | 18.29 | 17.65 | 50.00 | 79.07 | 31.54 | 0.57 | 48.50 | 47.64 | 49.20 | 31.57 |
| PyramidKV | 12.51 | 28.02 | 42.24 | 28.18 | 23.33 | 6.66 | 16.82 | 17.94 | 17.11 | 46.50 | 77.94 | 30.94 | 2.00 | 46.00 | 45.21 | 48.35 | 30.61 |
| CAKE | **14.09** | **30.33** | **44.33** | 28.63 | 24.47 | 6.70 | **18.20** | 18.83 | 18.98 | **53.00** | 79.54 | 32.14 | 1.50 | 48.00 | **48.71** | **50.59** | **32.38** |
| Gemma-7B-Instruct, $B_{\text{total}} = 1024L$ | | | | | | | | | | | | | | | | | |
| StreamingLLM | 13.23 | 27.04 | 29.08 | 25.39 | 22.15 | 5.67 | 20.94 | 17.74 | 21.97 | 67.50 | 79.28 | 33.64 | 1.50 | 11.50 | **49.80** | **49.25** | 29.73 |
| H2O | 13.39 | 30.25 | 41.23 | 28.75 | 24.41 | 7.31 | 22.85 | 18.68 | 22.73 | 66.50 | 79.78 | 33.12 | 1.57 | 44.50 | 49.49 | 48.55 | 33.32 |
| TOVA | 13.58 | 32.92 | 45.60 | 28.97 | 24.50 | 7.96 | 22.19 | 18.95 | 22.86 | **73.00** | 79.94 | 32.61 | 1.50 | 44.50 | 48.74 | 47.29 | 34.07 |
| SnapKV | 13.72 | 33.76 | 46.00 | 29.19 | 23.85 | 7.49 | 22.44 | 19.35 | 22.73 | 71.50 | **79.96** | 33.35 | 1.50 | 43.50 | 48.64 | 47.86 | 34.05 |
| PyramidKV | 14.32 | 33.24 | 46.35 | 28.75 | 23.51 | 6.91 | 21.66 | 19.37 | 22.47 | 66.00 | 78.95 | 33.52 | 1.00 | 43.50 | 47.43 | 48.07 | 33.44 |
| CAKE | **15.16** | **35.12** | **47.67** | 28.24 | 23.12 | 6.59 | **23.28** | 19.72 | **23.40** | 68.50 | 79.47 | **34.07** | 1.50 | 43.00 | 48.84 | 49.20 | **34.18** |

## F.2 Extended Evaluation on Additional Model Architectures

To demonstrate the generalizability of CAKE across different model architectures, we conduct additional experiments on Qwen2.5-7B-Instruct and Gemma-7B-Instruct. The experiments are conducted on two memory scenarios: a low setting ($B_{\text{total}} = 128L$) and a high setting ($B_{\text{total}} = 1024L$). As shown in Table 8, similar to results on Llama and Mistral, CAKE consistently outperforms baseline methods across both low-memory and high-memory scenarios on Qwen and Gemma architectures. CAKE demonstrates significant advantages in low-memory settings by rationally allocating cache sizes and evicting KV pairs with more robust indicators. Under the high-budget setting, CAKE achieves superior performance, even exceeding the full-cache baseline on Gemma (34.18 vs. 34.09). These results further validate CAKE's adaptability across different model architectures, maintaining its performance advantages regardless of the underlying model design.

Table 9: Performance comparison over 16 datasets of LongBench on different models with sizes from 13B to 70B. The best result is highlighted in **bold**, the second best in underline.

| Method | Single-Document QA | | | Multi-Document QA | | | Summarization | | | Few-shot Learning | | | Synthetic | | Code | | Avg. |
|---|---|---|---|---|---|---|---|---|---|---|---|---|---|---|---|---|---|
| | NrtvQA | Qasper | MF-en | HotpotQA | 2WikiMQA | Musique | GovReport | QMSum | MultiNews | TREC | TriviaQA | SAMSum | PCount | PR-en | Lcc | RB-P | |
| **Llama2-13B-Chat, $B_{total} = Full$** | | | | | | | | | | | | | | | | | |
| Full | 14.39 | 17.07 | 27.52 | 12.66 | 13.21 | 5.01 | 27.52 | 20.89 | 26.61 | 68.5 | 87.75 | 42.44 | 2.27 | 15.25 | 48.27 | 49.87 | 29.95 |
| **Llama2-13B-Chat, $B_{total} = 128L$** | | | | | | | | | | | | | | | | | |
| StreamingLLM | 10.48 | 13.44 | 19.16 | 12.86 | 13.81 | **4.90** | 15.63 | 18.91 | 16.26 | 40.00 | 81.66 | 34.00 | 2.29 | 5.00 | 40.38 | 35.16 | 22.75 |
| H2O | 12.74 | **17.63** | 25.71 | 14.12 | 13.29 | 3.84 | 19.35 | 19.71 | **21.51** | 40.00 | 80.62 | 39.37 | 2.03 | 10.50 | 40.75 | 41.31 | 25.15 |
| TOVA | 11.18 | 13.55 | 19.65 | 13.61 | 13.20 | 3.95 | 17.46 | 18.17 | 16.51 | **58.00** | **88.77** | 38.91 | 2.36 | 7.50 | 40.10 | 37.21 | 25.01 |
| SnapKV | **13.33** | 14.73 | 24.34 | 13.47 | 15.10 | 4.68 | **20.37** | 20.01 | 20.44 | 42.00 | 86.71 | 38.35 | **3.21** | 11.50 | 43.07 | 42.28 | 25.85 |
| PyramidKV | 13.04 | 13.91 | 25.26 | **14.37** | **15.24** | 4.68 | 20.01 | 19.74 | 20.52 | 43.50 | 85.37 | 38.83 | 2.80 | **13.00** | 43.57 | 41.31 | 25.95 |
| CAKE | 12.19 | 17.17 | **27.28** | 13.12 | 14.73 | 4.33 | 20.04 | **20.14** | 20.74 | 42.00 | 87.75 | **40.60** | 2.96 | 12.50 | **45.52** | **43.95** | **26.56** |
| **Llama2-13B-Chat, $B_{total} = 1024L$** | | | | | | | | | | | | | | | | | |
| StreamingLLM | 12.47 | 15.03 | 19.85 | 11.39 | 13.68 | 3.90 | 23.48 | 19.33 | 24.88 | 63.00 | 86.37 | 40.12 | **4.40** | 6.50 | 47.24 | 45.56 | 27.32 |
| H2O | 14.76 | **19.89** | **29.10** | **14.84** | 14.05 | 4.41 | 23.38 | 20.18 | 25.03 | 62.50 | 82.14 | 41.85 | 1.13 | 12.50 | 47.34 | 47.07 | 28.76 |
| TOVA | 13.69 | 16.10 | 24.41 | 12.79 | 13.43 | 4.96 | 23.81 | 20.22 | 24.99 | **69.00** | 88.15 | 42.18 | 3.19 | 13.25 | 46.71 | 48.77 | 29.10 |
| SnapKV | 14.39 | 17.87 | 27.99 | 12.94 | 13.51 | 4.95 | 24.65 | 20.74 | 25.08 | 68.50 | 86.47 | 42.26 | 2.62 | 14.00 | 48.27 | **49.42** | 29.60 |
| PyramidKV | 14.15 | 18.30 | 28.35 | 12.98 | 14.42 | 5.10 | **25.05** | **20.84** | **25.62** | 68.50 | **88.21** | 42.52 | 2.28 | 15.00 | 47.99 | 48.12 | 29.84 |
| CAKE | **15.33** | 17.84 | 29.05 | 13.99 | **14.30** | **5.46** | 24.43 | 20.60 | 25.29 | 68.00 | 87.42 | **42.58** | 3.14 | **15.50** | **48.40** | 48.41 | **29.98** |
| **Qwen2.5-32B-Instruct, $B_{total} = Full$** | | | | | | | | | | | | | | | | | |
| Full | 29.25 | 45.24 | 51.9 | 63.31 | 61.5 | 38.15 | 30.33 | 23.38 | 23.01 | 73.5 | 87.86 | 45.36 | 12.0 | 100.0 | 51.3 | 38.17 | 48.39 |
| **Qwen2.5-32B-Instruct, $B_{total} = 128L$** | | | | | | | | | | | | | | | | | |
| StreamingLLM | 16.26 | 22.62 | 25.95 | 44.14 | 47.57 | 23.37 | 15.66 | 17.50 | 13.22 | 41.50 | 81.47 | 38.03 | 12.00 | 58.04 | 43.51 | 32.46 | 33.33 |
| H2O | 24.03 | 24.48 | 31.10 | 52.53 | 52.38 | 30.84 | 17.00 | 19.67 | **16.27** | 43.00 | 83.64 | **41.41** | 12.22 | 94.58 | 45.85 | **34.35** | 38.96 |
| TOVA | **25.53** | 27.73 | 38.07 | 57.08 | 54.92 | 32.92 | **19.28** | 17.47 | 15.08 | **60.50** | 77.55 | 41.02 | 11.81 | **97.58** | 38.05 | 26.75 | 40.08 |
| SnapKV | 21.28 | 26.68 | 40.43 | 56.74 | 55.61 | **33.75** | 17.15 | 19.65 | 14.55 | 48.50 | **86.46** | 40.86 | **12.50** | 95.58 | 45.75 | 33.55 | 40.56 |
| PyramidKV | 18.31 | 24.62 | 36.85 | 55.30 | 55.40 | 32.38 | 15.33 | 18.02 | 13.15 | 46.50 | 84.50 | 39.62 | 11.50 | 88.83 | 44.05 | 31.76 | 38.51 |
| CAKE | 24.03 | **29.35** | **41.64** | **57.42** | **57.37** | 31.85 | 18.54 | **20.42** | 16.26 | 50.00 | 82.90 | 40.63 | **12.50** | 97.42 | **46.59** | 33.86 | **41.30** |
| **Qwen2.5-32B-Instruct, $B_{total} = 1024L$** | | | | | | | | | | | | | | | | | |
| StreamingLLM | 21.42 | 29.11 | 31.07 | 46.44 | 47.31 | 26.17 | 23.76 | 19.07 | 21.68 | 63.50 | 86.12 | 42.61 | **12.50** | 63.08 | 49.84 | 36.13 | 38.74 |
| H2O | 25.96 | 32.84 | 40.35 | 56.46 | 55.57 | 32.91 | 25.06 | 21.02 | 21.94 | 60.50 | 86.63 | 44.85 | **12.50** | 97.58 | 50.96 | 37.15 | 43.89 |
| TOVA | 29.38 | 40.61 | 48.18 | 59.65 | 60.91 | 34.77 | 24.86 | 20.69 | 22.03 | **73.00** | 87.34 | **45.09** | 12.00 | **100.00** | 49.11 | 35.90 | 46.47 |
| SnapKV | 29.93 | 42.29 | **50.00** | 61.97 | 61.37 | 37.22 | 25.01 | 22.47 | 21.94 | 69.50 | 87.31 | 44.42 | **12.50** | **100.00** | **51.54** | 37.32 | 47.17 |
| PyramidKV | 29.39 | 41.56 | 49.31 | **63.12** | 61.52 | **38.25** | 22.47 | 21.87 | 18.36 | 70.50 | **87.78** | 44.01 | 11.83 | **100.00** | 50.27 | 37.38 | 46.73 |
| CAKE | **30.77** | **44.49** | 50.73 | 61.59 | 60.60 | 37.39 | **25.86** | **22.77** | **22.18** | 72.50 | **87.78** | 43.84 | 12.12 | **100.00** | 51.13 | **37.71** | **47.59** |
| **Llama3-70B-Instruct, $B_{total} = Full$** | | | | | | | | | | | | | | | | | |
| Full | 26.08 | 47.26 | 49.73 | 49.61 | 54.84 | 27.51 | 31.01 | 22.48 | 27.99 | 73.5 | 92.88 | 45.21 | 11.0 | 68.5 | 37.72 | 67.28 | 45.79 |
| **Llama3-70B-Instruct, $B_{total} = 128L$** | | | | | | | | | | | | | | | | | |
| StreamingLLM | 22.33 | 28.13 | 28.07 | 42.85 | 50.73 | 22.18 | 17.75 | 20.02 | 18.23 | 44.00 | 82.95 | 40.05 | **11.50** | 67.50 | 44.44 | 59.76 | 37.53 |
| H2O | 24.92 | 29.15 | 36.50 | 45.98 | 50.40 | 24.52 | 19.38 | 20.61 | 21.40 | 43.50 | **91.78** | 42.90 | 10.55 | **70.15** | 46.92 | 63.45 | 40.13 |
| TOVA | 20.61 | 25.79 | 30.59 | 43.67 | 47.05 | 24.33 | 6.18 | 17.45 | 18.22 | **58.50** | 91.52 | 37.04 | 7.46 | 34.62 | 41.71 | 51.35 | 34.76 |
| SnapKV | 25.29 | 33.37 | 44.25 | 48.01 | 52.93 | **27.01** | 18.90 | 21.53 | 20.80 | 48.50 | 91.63 | 41.53 | 7.50 | 68.00 | 45.87 | **64.16** | 41.20 |
| PyramidKV | **25.87** | 34.16 | 41.26 | **48.55** | 52.24 | 24.19 | 19.17 | 21.20 | 21.02 | 46.50 | 91.43 | 41.90 | 9.00 | 68.00 | 46.52 | 63.02 | 40.88 |
| CAKE | 25.21 | **39.26** | **44.39** | 47.47 | **54.48** | 27.22 | 21.26 | **22.37** | **22.25** | 57.00 | 91.28 | **43.69** | 10.00 | 67.50 | 44.41 | **64.16** | **42.62** |
| **Llama3-70B-Instruct, $B_{total} = 1024L$** | | | | | | | | | | | | | | | | | |
| StreamingLLM | 25.18 | 35.93 | 32.89 | 45.57 | 49.49 | 21.63 | 24.42 | 20.28 | 26.03 | 66.00 | 91.32 | 43.53 | **11.00** | 68.00 | 42.61 | 65.87 | 41.86 |
| H2O | 26.66 | 41.54 | 44.56 | 47.47 | 53.84 | 26.26 | 26.05 | 21.52 | 26.75 | 67.00 | 92.88 | 44.90 | **11.00** | 68.00 | 40.43 | **69.09** | 44.25 |
| TOVA | **27.03** | 45.73 | 47.61 | **49.77** | 55.54 | 27.02 | 26.04 | 21.59 | 26.58 | 73.00 | **92.95** | **46.24** | **11.00** | 68.00 | 39.76 | 62.59 | 45.03 |
| SnapKV | 26.63 | 45.96 | 47.66 | 48.57 | 54.92 | **27.60** | 26.31 | 22.34 | 26.75 | 72.00 | 92.88 | 45.10 | 10.00 | 68.50 | 40.72 | 67.22 | 45.20 |
| PyramidKV | 26.99 | 45.44 | 48.01 | 49.04 | 54.17 | 27.37 | 26.08 | **22.45** | 26.22 | 73.00 | 91.95 | 45.10 | 10.50 | 68.50 | 38.44 | 67.95 | 45.08 |
| CAKE | 26.95 | **46.94** | **48.51** | 47.91 | **55.98** | 27.46 | **26.39** | 22.44 | **27.44** | **73.50** | 92.44 | 45.60 | **11.00** | 68.00 | **44.23** | 68.47 | **45.83** |

## F.3 EXTENDED EVALUATION ON LARGER-SCALE MODELS

We further extend our experiments to models ranging from 13B to 70B parameters, including Llama2-13B-Chat, Qwen2.5-32B-Instruct, and Llama3-70B-Instruct, under two settings: $B_{total} = 128L$ and $B_{total} = 1024L$. In Table 9, CAKE outperforms baseline methods across all model sizes. Notably, under the high memory setting ($B_{total} = 1024L$), CAKE achieves even better performance than full-cache settings for both Llama2-13B (29.98 *vs.* 29.95) and Llama3-70B (45.83 *vs.* 45.79), showing that our method scales well to larger models while maintaining its efficiency advantages.

## G    Extended Experimental Results on NeedleBench

In this section, we provide more detailed experimental results on NeedleBench (Li et al., 2024a), using Mistral-7B-Instruct-v0.3 (Jiang et al., 2023) and Llama3.1-8B-Instruct (Dubey et al., 2024) as backbone LLMs. We examine three distinct tasks on NeedleBench: (1) *Single-Needle Retrieval (S-RT):* Assessing precision in locating individual details within extensive texts. (2) *Multi-Needle Retrieval (M-RT):* Evaluating the retrieval of multiple related information pieces dispersed throughout lengthy texts. (3) *Multi-Needle Reasoning (M-RS):* Measuring complex reasoning abilities by extracting multiple information elements from extended contexts to answer questions.

All experiments are conducted rigorously following the original NeedleBench protocol: Levenshtein distance is used to measure the similarity between predictions and references. Each case is repeated ten times to ensure stable scores, and the results are weighted-averaged to obtain an overall score, providing a balanced representation of each task.

**Results on Mistral-7B-Instruct-v0.3**. The results presented in Table 10 and Table 11 demonstrate that CAKE consistently outperforms other KV cache eviction methods across all evaluated tasks. CAKE exhibits the smallest decrease in performance scores when compared with the full cache baseline. The robustness of CAKE is evident across all tested cache sizes, with particularly notable improvements in Multi-Needle Retrieval tasks (Figure 10 and Figure 11). This superior performance indicates CAKE's enhanced ability to tolerate attention-shifts in long contexts, which can be attributed to its eviction strategy that incorporates both sustained importance and attention variability. The proposed indicator and strategy used in CAKE proves to be effective in maintaining performance even under constrained cache conditions.

**Results on Llama3.1-8B-Instruct**. We further evaluate CAKE's performance on Llama3.1-8B-Instruct using NeedleBench 32K. The results presented in Table 12 illustrate CAKE's comprehensive advantages across all three tasks, and Figure 12 showcases CAKE's outstanding performance on multibench tasks, confirming its effectiveness on models with GQA architecture as well.

On the whole, CAKE achieves better overall performance across the two models and various cache sizes, especially for Multi-Needle Retrieval tasks. Yet, given extremely small cache size budgets, Multi-Needle Retrieval tasks performance experiences a sharp decline in performance. This may be attributed to the stringent cache limitations are inadequate to support the information demanded by multiple needles. Nevertheless, compared to alternative approaches, CAKE demonstrates superior capability in mitigating this issue. Future research should focus on exploring more sophisticated methods for retaining KV pairs, as this remains a promising avenue for further improvement.

Table 10: Performance comparison over three subtasks of NeedleBench 32K benchmark on Mistral-7B-Instruct-v0.3. The best is highlighted in **bold**, the second best is in underline.

| Method | Single-Retrieval | | | Multi-Retrieval | | | Multi-Reasoning | | | Overall |
| --- | --- | --- | --- | --- | --- | --- | --- | --- | --- | --- |
| | ZH | EN | Overall | ZH | EN | Overall | ZH | EN | Overall | |
| *Mistral-7B-Instruct-v0.3, $B_{\text{total}} = Full$* | | | | | | | | | | |
| Full | 91.72 | 83.01 | 87.37 | 90.36 | 96.95 | 93.66 | 55.64 | 45.89 | 50.77 | 78.28 |
| *Mistral-7B-Instruct-v0.3, $B_{\text{total}} = 1024L$* | | | | | | | | | | |
| H2O | 59.20 | 39.75 | 49.48 | 39.50 | 32.91 | 36.20 | 36.95 | **46.58** | 41.76 | 43.18 |
| SnapKV | 91.73 | 90.25 | 90.99 | 32.27 | 81.59 | 56.93 | 48.87 | 44.34 | 46.61 | 67.46 |
| PyramidKV | **91.94** | **90.69** | **91.32** | 25.23 | 70.55 | 47.89 | 47.68 | 45.32 | 46.50 | 64.84 |
| CAKE | 89.43 | 89.89 | 89.66 | **51.68** | **90.32** | **71.00** | **52.50** | 45.57 | **49.04** | **71.87** |
| *Mistral-7B-Instruct-v0.3, $B_{\text{total}} = 512L$* | | | | | | | | | | |
| H2O | 60.87 | 38.44 | 49.66 | 4.50 | 12.68 | 8.59 | 36.21 | 46.15 | 41.18 | 34.79 |
| SnapKV | **91.21** | 90.49 | 90.85 | 4.95 | 33.32 | 19.14 | 43.88 | 45.20 | 44.54 | 55.44 |
| PyramidKV | 90.06 | 88.42 | 89.24 | 3.50 | 17.91 | 10.70 | 42.43 | 45.77 | 44.10 | 52.14 |
| CAKE | 89.97 | **92.06** | **91.01** | **26.18** | **70.91** | **48.55** | **47.60** | **46.28** | **46.94** | **65.05** |
| *Mistral-7B-Instruct-v0.3, $B_{\text{total}} = 256L$* | | | | | | | | | | |
| H2O | 66.54 | 40.69 | 53.61 | 0.23 | 4.64 | 2.43 | 34.61 | **47.74** | 41.17 | 34.53 |
| SnapKV | 89.27 | 89.10 | 89.18 | 0.59 | 2.95 | 1.77 | 35.67 | 47.10 | 41.39 | 48.62 |
| PyramidKV | 81.51 | 84.87 | 83.19 | 0.18 | 4.27 | 2.23 | 33.41 | 46.90 | 40.16 | 45.99 |
| CAKE | **90.33** | **89.83** | **90.08** | **1.27** | **7.50** | **4.39** | **39.03** | 47.21 | **43.12** | **50.29** |

Table 11: Performance comparison over three subtasks of NeedleBench 8K benchmark on Mistral-7B-Instruct-v0.3. The best is highlighted in **bold**, the second best is in underline.

| Method | Single-Retrieval | | | Multi-Retrieval | | | Multi-Reasoning | | | Overall |
|---|---|---|---|---|---|---|---|---|---|---|
| | ZH | EN | Overall | ZH | EN | Overall | ZH | EN | Overall | |
| Mistral-7B-Instruct-v0.3, $B_{\text{total}} = Full$ | | | | | | | | | | |
| Full | 98.33 | 84.96 | 91.65 | 98.25 | 96.85 | 97.55 | 63.59 | 57.14 | 60.37 | 84.03 |
| Mistral-7B-Instruct-v0.3, $B_{\text{total}} = 1024L$ | | | | | | | | | | |
| H2O | 77.93 | 54.86 | 66.39 | 5.15 | 42.20 | 23.67 | 43.55 | 57.39 | 50.47 | 48.80 |
| SnapKV | 97.66 | 89.92 | 93.79 | 12.25 | 50.65 | 31.45 | 57.38 | 57.52 | 57.45 | 64.19 |
| PyramidKV | **98.68** | 87.55 | 93.11 | 9.95 | 37.65 | 23.80 | 55.59 | 56.77 | 56.18 | 61.24 |
| CAKE | 98.20 | 90.27 | **94.24** | **53.80** | **87.25** | **70.53** | **57.75** | **57.57** | **57.66** | **76.15** |
| Mistral-7B-Instruct-v0.3, $B_{\text{total}} = 512L$ | | | | | | | | | | |
| H2O | 72.73 | 50.71 | 61.72 | 0.85 | 2.75 | 1.80 | 39.79 | 57.86 | 48.82 | 39.87 |
| SnapKV | **98.11** | 90.58 | 94.35 | 2.25 | 3.85 | 3.05 | 52.71 | 57.78 | 55.25 | 55.23 |
| PyramidKV | 97.45 | 87.87 | 92.66 | 1.25 | 3.00 | 2.12 | 49.23 | **58.52** | 53.87 | 53.86 |
| CAKE | 98.08 | **92.33** | **95.20** | **16.60** | **45.25** | **30.93** | **52.78** | 58.48 | **55.63** | **64.05** |
| Mistral-7B-Instruct-v0.3, $B_{\text{total}} = 256L$ | | | | | | | | | | |
| H2O | 74.35 | 48.40 | 61.37 | 0.20 | 0.20 | 0.20 | 37.38 | 57.87 | 47.62 | 38.90 |
| SnapKV | **97.47** | 91.89 | **94.68** | 0.60 | 0.50 | 0.55 | 42.28 | 57.93 | 50.10 | 53.07 |
| PyramidKV | 92.02 | 88.24 | 90.13 | 0.30 | 0.55 | 0.43 | 39.88 | 57.02 | 48.45 | 50.71 |
| CAKE | 97.22 | 91.93 | 94.57 | **1.00** | **1.45** | **1.22** | **44.21** | **58.73** | **51.47** | **53.64** |

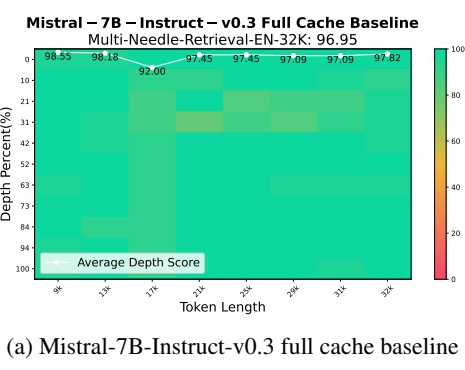

(a) Mistral-7B-Instruct-v0.3 full cache baseline

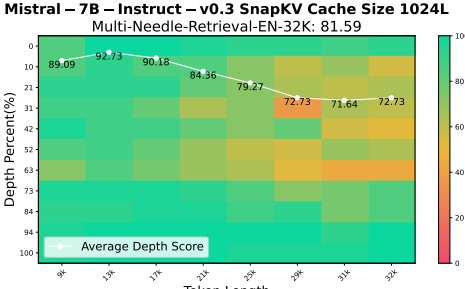

(b) SnapKV: cache budget $B_{\text{total}} = 1024L$

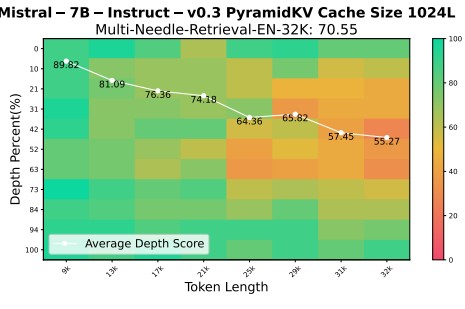

(c) PyramidKV: cache budget $B_{\text{total}} = 1024L$

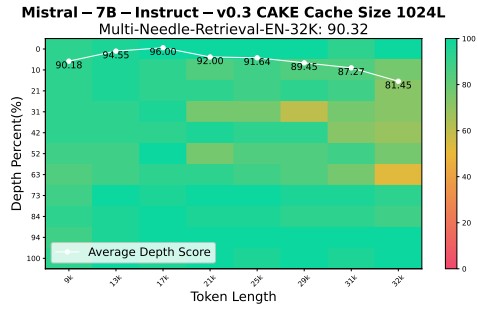

(d) CAKE: cache budget $B_{\text{total}} = 1024L$

Figure 10: Performance comparison of Mistral-7B-Instruct-v0.3 on NeedleBench 32K Multi-Needle Retrieval Task (EN) using different KV eviction methods under $B_{\text{total}} = 1024L$.

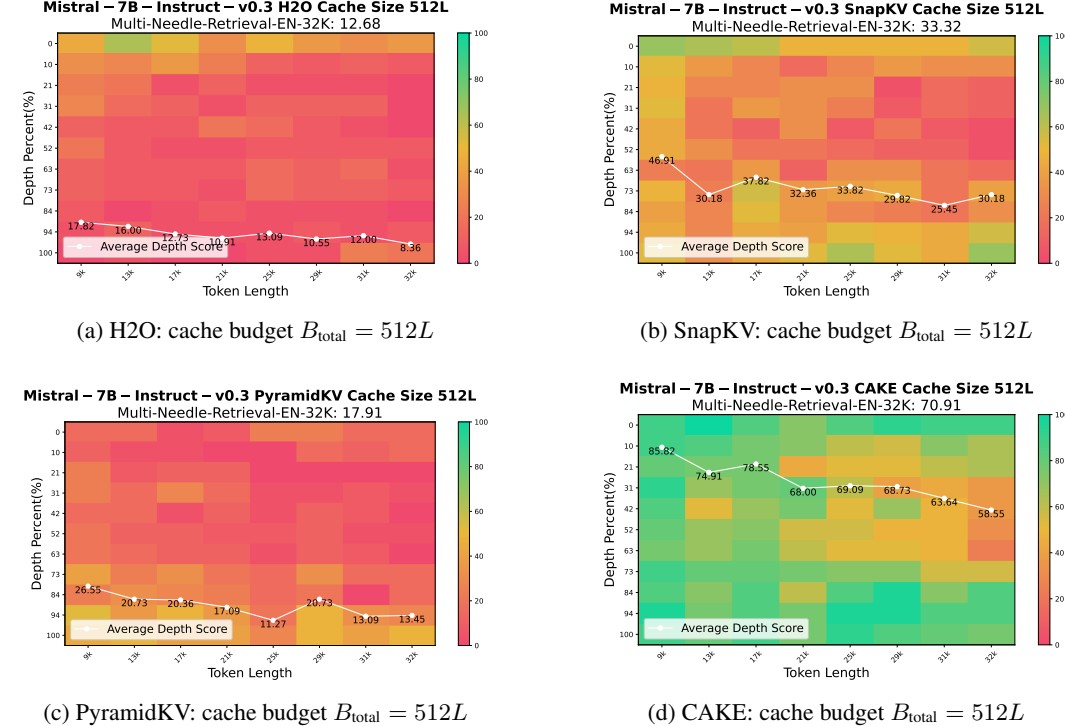

(a) H2O: cache budget $B_{\text{total}} = 512L$

(b) SnapKV: cache budget $B_{\text{total}} = 512L$

(c) PyramidKV: cache budget $B_{\text{total}} = 512L$

(d) CAKE: cache budget $B_{\text{total}} = 512L$

Figure 11: Performance comparison of Mistral-7B-Instruct-v0.3 on NeedleBench 32K Multi-Needle Retrieval Task (EN) using different KV eviction methods under $B_{\text{total}} = 512L$.

Table 12: Performance comparison over three subtasks of NeedleBench 32K benchmark on Llama3.1-8B Instruct. The best is highlighted in **bold**, the second best is in underline.

| Method | Single-Retrieval | | | Multi-Retrieval | | | Multi-Reasoning | | | Overall |
|---|---|---|---|---|---|---|---|---|---|---|
| | ZH | EN | Overall | ZH | EN | Overall | ZH | EN | Overall | |
| Llama3.1-8B Instruct, $B_{\text{total}} = Full$ | | | | | | | | | | |
| Full | 100.00 | 82.99 | 91.49 | 99.64 | 99.77 | 99.70 | 72.50 | 77.70 | 75.10 | 89.04 |
| Llama3.1-8B Instruct, $B_{\text{total}} = 1024L$ | | | | | | | | | | |
| H2O | 89.51 | 66.59 | 78.05 | **89.45** | 91.77 | 90.61 | 54.52 | 71.91 | 63.21 | 77.37 |
| SnapKV | 99.90 | 84.21 | 92.06 | 86.68 | 98.23 | 92.45 | 71.11 | 73.29 | 72.20 | 86.22 |
| PyramidKV | **100.00** | **85.53** | **92.76** | 56.77 | 88.55 | 72.66 | **71.59** | **76.16** | **73.87** | 81.07 |
| CAKE | **100.00** | 84.02 | 92.01 | 89.41 | **98.91** | **94.16** | 69.29 | 73.42 | 71.35 | **86.46** |
| Llama3.1-8B Instruct, $B_{\text{total}} = 512L$ | | | | | | | | | | |
| H2O | 88.14 | 66.08 | 77.11 | 30.05 | 80.14 | 55.09 | 50.97 | 73.20 | 62.08 | 66.00 |
| SnapKV | 99.90 | 84.30 | 92.10 | 36.09 | 78.05 | 57.07 | **69.20** | 73.41 | 71.31 | 75.35 |
| PyramidKV | **100.00** | **86.17** | **93.09** | 11.00 | 49.36 | 30.18 | 68.72 | **77.13** | **72.93** | 68.17 |
| CAKE | **100.00** | 82.07 | 91.04 | **47.36** | **85.68** | **66.52** | 67.67 | 71.15 | 69.41 | **77.19** |
| Llama3.1-8B Instruct, $B_{\text{total}} = 256L$ | | | | | | | | | | |
| H2O | 88.54 | 69.61 | 79.08 | 4.95 | 6.23 | 5.59 | 50.11 | 72.86 | 61.49 | 51.75 |
| SnapKV | 99.90 | 82.90 | 91.40 | 4.05 | 8.45 | 6.25 | **66.68** | 74.48 | **70.58** | 59.61 |
| PyramidKV | **100.00** | **85.07** | **92.53** | 1.91 | 6.09 | 4.00 | 64.55 | **76.14** | 70.34 | 59.32 |
| CAKE | 99.81 | 81.92 | 90.86 | **6.36** | **27.64** | **17.00** | 64.57 | 68.61 | 66.59 | **61.42** |

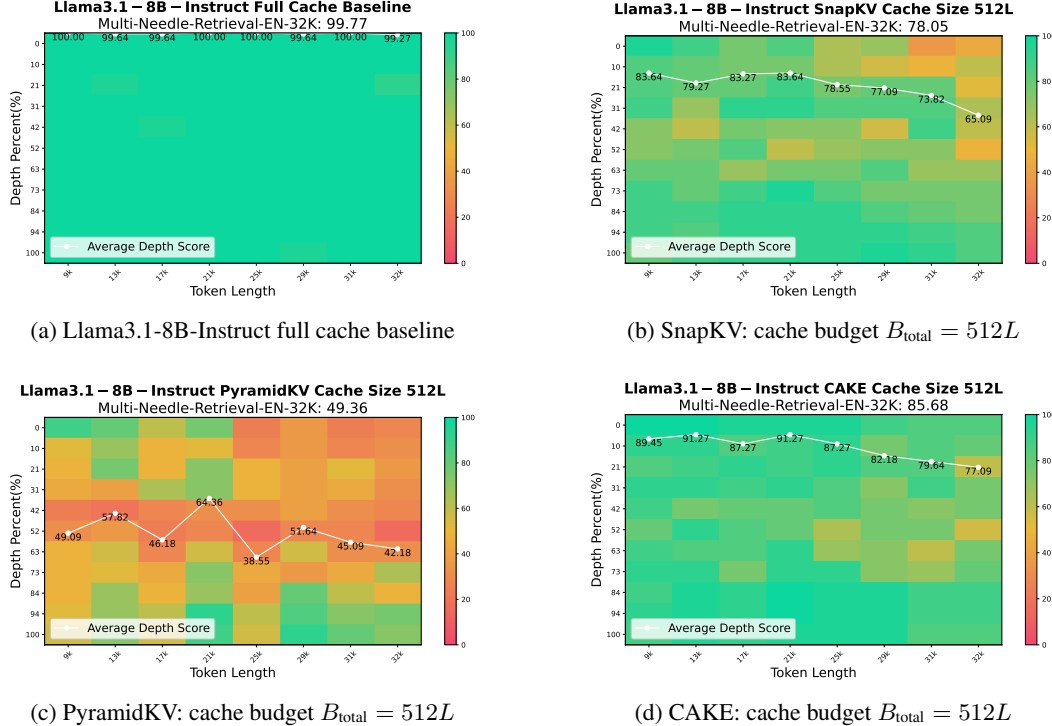

Figure 12: Performance comparison of Llama3.1-8B-Instruct on NeedleBench 32K Multi-Needle Retrieval Task (EN) using different KV eviction methods under $B_{\text{total}} = 512L$.

Table 13: Prompt prefilling and token decoding latency comparison.

| Prefill+Decoding Length | Method | Overall Generation Time (s) | Prompt Prefill Time (s) | Decoding Time (s) | Decoding Time per Token (ms) |
|---|---|---|---|---|---|
| 7168+1024 | SnapKV | 33.74 | 0.71 | 33.03 | 32.25 |
| | PyramidKV | 32.29 | 0.78 | 31.50 | 30.77 |
| | CAKE | 31.77 | 0.81 | 30.96 | 30.24 |
| | Full Cache | 34.54 | 0.70 | 33.83 | 33.04 |
| 15360+1024 | SnapKV | 34.78 | 1.66 | 33.12 | 32.34 |
| | PyramidKV | 33.02 | 1.73 | 31.29 | 30.56 |
| | CAKE | 32.96 | 1.77 | 31.19 | 30.46 |
| | Full Cache | 50.71 | 1.66 | 49.06 | 47.91 |
| 31744+1024 | SnapKV | 37.27 | 4.13 | 33.15 | 32.37 |
| | PyramidKV | 35.78 | 4.20 | 31.58 | 30.84 |
| | CAKE | 36.42 | 4.29 | 32.12 | 31.37 |
| | Full Cache | 86.42 | 4.14 | 82.28 | 80.35 |
| 64512+1024 | SnapKV | 44.29 | 11.33 | 32.96 | 32.19 |
| | PyramidKV | 43.11 | 11.48 | 31.63 | 30.89 |
| | CAKE | 43.10 | 11.47 | 31.63 | 30.89 |
| | Full Cache | 176.95 | 11.42 | 165.53 | 161.65 |
| 130048+1024 | SnapKV | 68.98 | 35.38 | 33.60 | 32.81 |
| | PyramidKV | 67.61 | 35.51 | 32.10 | 31.35 |
| | CAKE | 66.83 | 35.55 | 31.28 | 30.54 |
| | Full Cache | 364.15 | 35.86 | 328.28 | 320.59 |

## H ADDITIONAL EXPERIMENTS AND ANALYSIS ON EFFICIENCY

In this section, we present a detailed time breakdown during both prompt prefilling and token decoding to better assess CAKE's effectiveness across different inference stages compared to other KV eviction methods, including SnapKV and PyramidKV with various allocation strategies. All methods are implemented as extensions of FlashAttention-2 (Dao, 2023) using Mistral-7B-Instruct-v0.3, and their performance was benchmarked against a full cache baseline utilizing FlashAttention. To measure computational time more precisely throughout the entire text generation phase, we have separately evaluated the prompt prefilling time and the duration of autoregressive decoding. By fixing the generation token count at 1024, the experiment simulates typical long-context processing scenarios in LLMs, which are characterized by lengthy inputs and concise outputs.

As shown in Table 13, while dramatically lowering latency in the decoding phase, CAKE, SnapKV, and PyramidKV achieve comparable prompt prefilling times. For SnapKV and PyramidKV with fixed allocation strategies, the prompt prefilling stage requires waiting for KV caching and performing one eviction operation per layer, resulting in $L$ eviction operations. However, compared to the computation-intensive prefill stage, the eviction operation is relatively inexpensive and can be considered negligible. Notably, while CAKE utilizes preference-guided cascading cache management, it achieves similar timing to SnapKV and PyramidKV with fixed allocation strategies. This is attributed to the parallelizable execution of KV cache eviction between layers (Algorithm 1, lines 9–14), which allows the dynamic update process to be incorporated into a single eviction operation. Therefore, the total execution time ultimately equates to that of performing $L$ eviction operations.

## I ADDITIONAL ABLATION STUDIES

### I.1 ABLATION STUDY ON METHOD COMPONENTS

In this section, we conduct ablation studies on LongBench to analyze the contribution of each component in our method. These studies highlight the importance of our design choices, focusing on two key aspects: (1) the preference-prioritized adaptive allocation strategy, which combines attention dispersion ($\mathcal{H}$) for spatial influence and attention shift ($\mathcal{V}$) for temporal dynamics, and (2) the attention-shift tolerant eviction indicator, which integrates mean attention score (Mean) for identifying sustained important tokens and attention variance (Var) to capture dynamic attention changes.

As shown in Table 14, incorporating each additional component leads to a noticeable performance improvement, demonstrating the synergistic effect of our method. The preference-prioritized adaptive allocation strategy improves performance from 28.14 to 28.82 when both $\mathcal{H}$ and $\mathcal{V}$ are included, underscoring the importance of capturing spatial and temporal attention dynamics. This highlights how our approach goes beyond traditional uniform strategies by tailoring cache allocation based on complex layer-specific attention patterns. Furthermore, incorporating attention variance (Var) in the eviction indicator significantly enhances performance by accounting for attention fluctuations, further improving average performance to 29.29. This confirms that addressing both spatial distribution and temporal evolution of attention is crucial for effective KV cache management, especially in long-context scenarios where dynamic attention patterns heavily influence performance. Our comprehensive approach successfully balances these factors, yielding superior results compared to methods that overlook such nuanced attention characteristics.

Table 14: Ablation study on different components of our method. The experiments are conducted on Llama2-7b-Chat with total cache budget $B_{\textbf{total}} = 128L$.

| Allocation strategy | | Eviction indicator | | Avg. |
|---|---|---|---|---|
| $\mathcal{H}$ | $\mathcal{V}$ | Mean | Var | |
| | | ✓ | | 28.07 |
| ✓ | | ✓ | | 28.14 |
| ✓ | ✓ | ✓ | | 28.82 |
| ✓ | ✓ | ✓ | ✓ | **29.29** |

## I.2 INFLUENCE OF TEMPERATURE PARAMETER

In this section, we examine the impact of temperature parameters $\tau_1$ and $\tau_2$, which modulate the influence of attention dispersion and shift in our preference-prioritized adaptive allocation strategy. We evaluated various settings on LLama3.1-8B-Instruct and Mistral-7B-Instruct-v0.3 models, using a cache budget of $B_{\text{total}} = 128L$. Our approach is compared against SnapKV, a state-of-the-art KV eviction method employing uniform cache allocation, which serves as our baseline. As shown in Figure 13, our method consistently outperforms the baseline across different configurations for both models. Mistral-7B-Instruct-v0.3 achieves a peak performance of 37.33 (baseline: 35.81), while LLama3.1-8B-Instruct reached 43.39 (baseline: 42.53), demonstrating the robustness and effectiveness of our strategy. While our approach is inherently robust, we found that simple, low-cost adjustments to $\tau_1$ and $\tau_2$ can further enhance performance. These adjustments allow for better capture of model-specific attention patterns, optimizing cache allocation without expensive retraining or extensive hyperparameter searches. This characteristic is particularly valuable in industrial deployments where specific models need to be optimized under constrained memory budgets, offering a practical solution for resource-efficient inference in production environments.

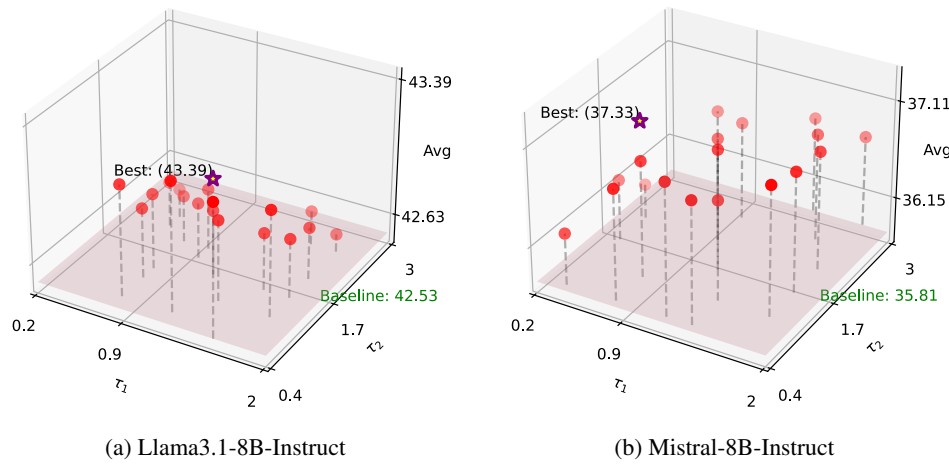

(a) Llama3.1-8B-Instruct           (b) Mistral-8B-Instruct

Figure 13: Performance comparison of different $\tau_1$ and $\tau_2$ configurations for Mistral-7B-Instruct-v0.3 and Llama3.1-8B-Instruct models. The plots show the impact of varying $\tau_1$ and $\tau_2$ on model performance, with the baseline (SnapKV) and best scores indicated.

## J ADDITIONAL DETAILED ANALYSIS AND VISUALIZATION OF ATTENTION DYNAMICS

In this section, we provide a more intuitive and comprehensive analysis of the differences in attention patterns by presenting detailed visualizations of attention weights across various dimensions. To illustrate the variations between different layers, we visualize the model's attention weights across all layers by aggregating the attention weights of each attention head. We also highlight the differences in attention between different models by conducting experiments on both Llama3 and Mistral. To demonstrate the attention differences across various input contexts, we visualize attention weights for different task types on Llama3, including single-document QA, summarization, and code tasks. Additionally, we visualize attention weights for the same task types but with varying contexts on Mistral.

**Attention Differences across Layers**. As shown in Figure 14 and Figure 15, attention patterns exhibit significant differences across layers, as mentioned in the main text. Regardless of the model or input context, we can observe varying degrees of attention dispersion and attention shift across different layers. For example, we see layers with higher attention dispersion (*e.g.*, layer 0 in Figure 14(b)) or lower dispersion (*e.g.*, layer 1 in Figure 14(b)), as well as layers with higher shift (*e.g.*, layer 24 in Figure 15 (c)) or lower shift (*e.g.*, layer 35 in Figure 15(c)). These observations demon-

strate the importance of carefully considering the differences in attention mechanisms and allocating appropriate cache resources accordingly.

**Attention Differences across Models**. As shown in Figure 14 and Figure 15, these models have notable differences in attention patterns. For instance, Mistral often exhibits higher attention dispersion at the beginning and end of its layer stack. In contrast, Llama3 frequently displays high attention dispersion primarily in the early layers. These distinct patterns underscore the importance of model-specific considerations when designing caching strategies, as different architectures may have varying attention characteristics across their layer stacks.

**Attention Differences across Contexts**. As evident in Figure 14(a)-(c), there are striking differences in attention patterns across different task types. Even within the same task type, while there may be some similarities in attention dispersion, attention shift patterns show marked differences. This can lead to different layer preferences for KV cache (comparing layers 10-16 in Figure 15(a) with layers 10-16 in Figure 15(b)). Therefore, whether the task types are the same or different, attention patterns are consistently dynamic and varied.

Given the dynamic nature of attention across layers, models, and even contexts, adopting a uniform allocation strategy, while safe, does not effectively utilize memory. On the other hand, using a fixed pattern allocation strategy fails to generalize effectively across multiple models and task scenarios. In contrast, our CAKE thoroughly considers layer preferences for KV cache, formulating a reasonable cache size allocation scheme from a global perspective. It takes into account the substantial impact of attention dynamics, allowing it to dynamically adapt to different models and contexts. This approach ensures that CAKE can effectively respond to the varied and changing attention patterns observed across different scenarios.

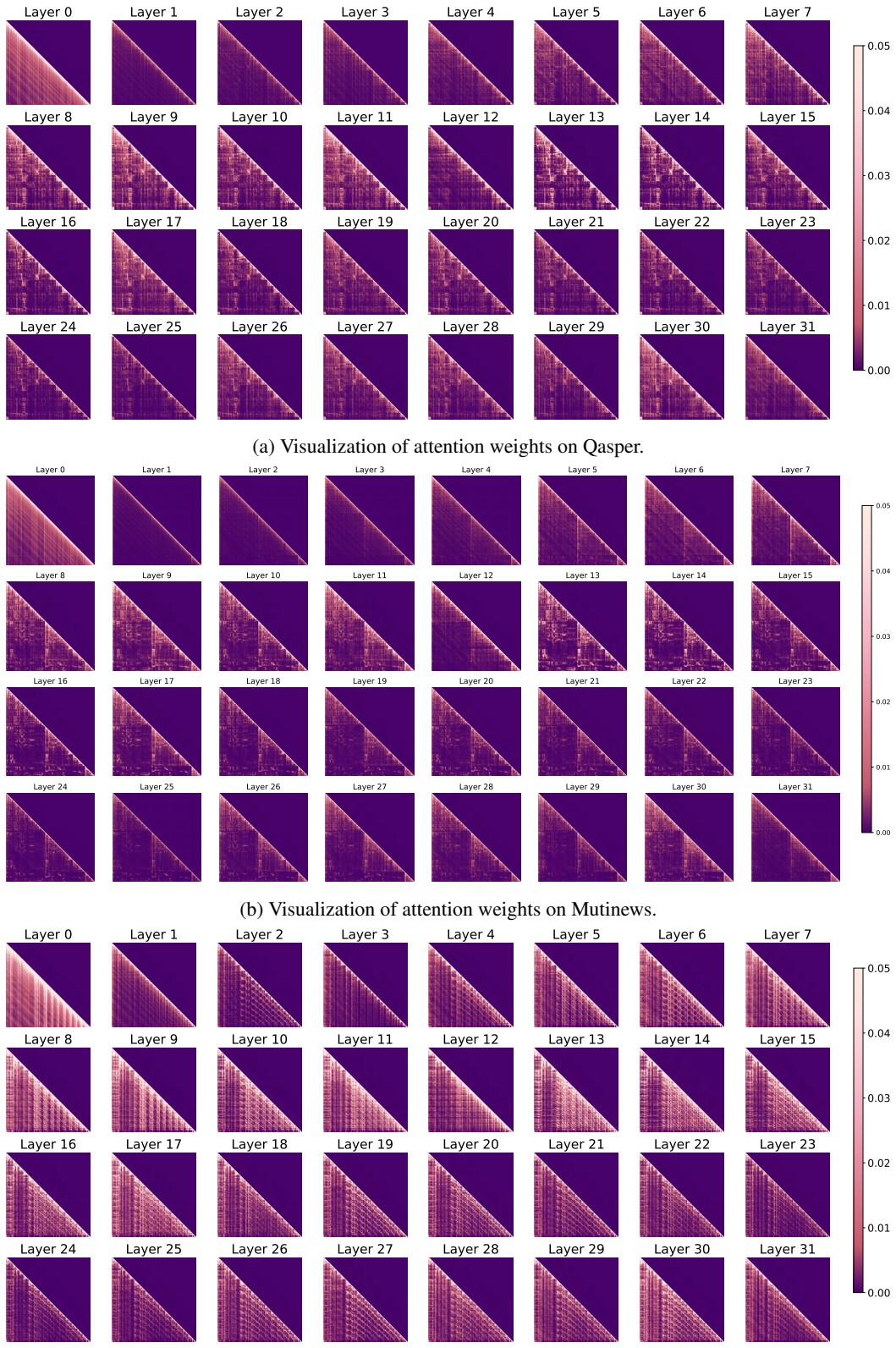

(a) Visualization of attention weights on Qasper.

(b) Visualization of attention weights on Mutinews.

(c) Visualization of attention weights on Lcc.

Figure 14: Visualization results of attention weights for Llama3.1-8B-Instruct across datasets from different tasks.

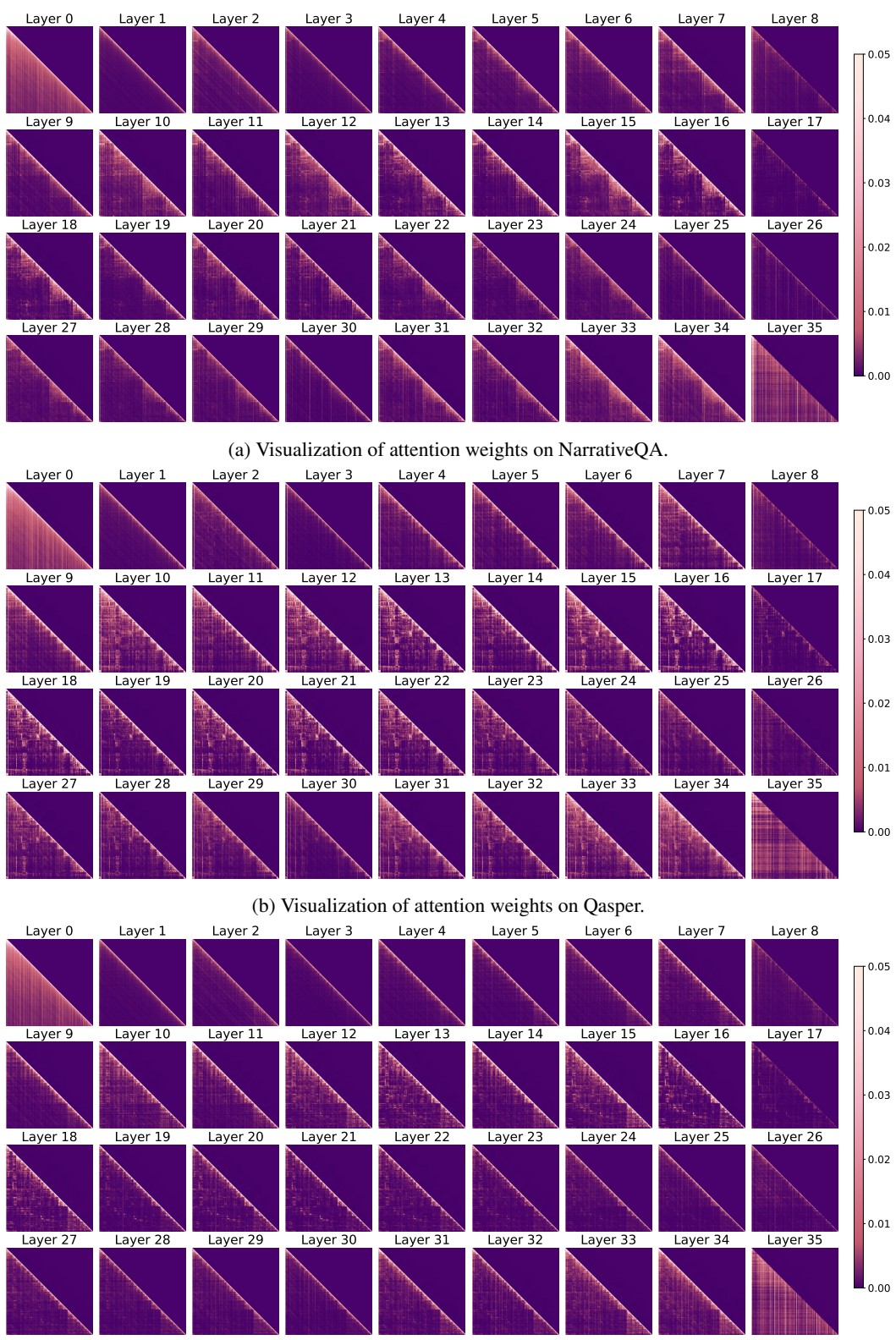

Figure 15: Visualization results of attention weights for Mistral-7B-Instruct-v0.3 across datasets from Single-document QA task.

