# OpenReview forum: "CAKE: Cascading and Adaptive KV Cache Eviction with Layer Preferences"
_ICLR.cc/2025/Conference — ICLR 2025 Poster_

### Official Review · Reviewer_zgXb · 2024-11-02

**Soundness:** 3
**Presentation:** 2
**Contribution:** 2
**Rating:** 6
**Confidence:** 4

**Summary:**

This work introduces CAKE, a method for efficient KV cache management in LLMs that enhances inference by dynamically allocating cache based on each layer’s spatial and temporal attention demands. By framing cache eviction as a cake-slicing problem, CAKE optimally distributes resources across layers and incorporates a novel eviction indicator to account for the shifting importance of tokens over time. Extensive experiments show the potentials of CAKE.

**Strengths:**

- Pioneering Adaptive Memory Budgets: CAKE is the first work to consider adaptive memory budgets for different layers of LLMs. This innovative approach allows for more efficient memory utilization, improving model performance by allocating resources where they are most needed.

- Addressing KV Cache Compression: The paper tackles the timely problem of KV cache compression in LLMs, which is especially relevant for on-device applications. By focusing on this issue, the work makes LLMs more practical and accessible in resource-constrained environments.

- Clear Writing: The overall writing is clear, though the paper structure is a bit chaotic. This clarity facilitates comprehension, replication, and further research based on the paper's findings.

**Weaknesses:**

- The motivation for using spatial dispersion and temporal shift in cache size allocation is unclear. Providing more insights or intuition would help clarify its benefits. Additionally, Table 2 shows that the adaptive allocation strategy provides minimal improvement, suggesting it may not be necessary.

- The adaptive KV compression method is incompatible with flash-attention. Given that flash-attention is widely adopted for efficient training and inference of LLMs, it’s unlikely that practitioners would choose CAKE over flash-attention in practice.

- Only two LLM backbones, Llama and Mistral, were evaluated, which may be insufficient. Consider adding another backbone, such as Phi, Qwen, or Gemma, to strengthen the analysis.

- The team has not open-sourced their code, which could raise concerns about the reproducibility of their work.

- The baseline methods used for comparison are not sufficiently strong. Consider including the following more robust methods (For KVQuant[1] and KIVI[2] should be jointly used with the flash-attention, and GEAR could remain the original settings):

[1] Hooper, Coleman, et al., "Kvquant: Towards 10 Million Context Length LLM Inference with KV Cache Quantization." arXiv preprint arXiv:2401.18079 (2024).

[2] Liu, Zirui, et al., "KIVI: Plug-and-Play 2bit KV Cache Quantization with Streaming Asymmetric Quantization." (2024).

[3] GEAR: An Efficient KV Cache Compression Recipe for Near-Lossless Generative Inference of LLM.

**Questions:**

Please refer to the weakness

---

> ### Author Response · Authors · 2024-11-23
> **Response to reviewer zgXb, part1**
>
> We greatly appreciate your thorough review and detailed suggestions. Addressing your comments has helped us strengthen both the technical content and clarity of our presentation. Below we address your concerns point by point. Corresponding modifications in the paper are **highlighted in blue**.
> >#### **W1**: The motivation for using spatial dispersion and temporal shift.
>
> Our approach is motivated by a key observation: attention patterns exhibit significant variations across layers, models, and contexts, as demonstrated by our extensive visualizations in **Appendix J**. This characteristic makes it suboptimal to employ fixed or uniform cache allocation strategies. Through our analysis, we identify two crucial aspects of attention patterns: spatial dispersion and temporal shift. Spatially, we observe that some layers distribute attention broadly across tokens (Fig. 1(a)), while others concentrate on specific tokens (Fig. 1(b)). Layers with broader attention dispersion, when compared with full cache settings, require larger cache sizes to maintain their performance. However, spatial patterns alone cannot fully capture attention dynamics. Temporally, we find that some layers shift their attention focus across different tokens during different steps (Fig. 1(c)), while others maintain fixed attention to tokens (Fig. 1(d)). Layers with dynamic temporal patterns need larger cache allocations to effectively track these changes. Given these observations, we propose that effective cache allocation must consider both spatial dispersion and temporal shift to accurately measure each layer's cache requirements. This comprehensive approach enables CAKE to adaptively allocate resources based on layer preferences, adapting to varying attention patterns in different models and contexts.
>
> Indeed, the above analyses have been provided in **67-75 lines** of the submitted paper. Please kindly refer to it. We trust that this explanation adequately addresses your concerns. Should you require additional clarification, please do not hesitate to inform us, as we are more than willing to provide it.
>
>
> >#### **W2**:  Table 2 shows that the adaptive allocation strategy provides minimal improvement.
>
> To better demonstrate the effectiveness of our adaptive allocation strategy, we have conducted additional experiments across multiple models, comparing three allocation strategies: uniform allocation, pyramid allocation, and our preference-prioritized adaptive allocation (P2A). The table below presents average scores across 16 LongBench datasets with a total budget size of 128L:
>
> | Model | Uniform | Pyramid | P2A (Ours) |
> |-------|----------|----------|------------|
> | LLama2-7B-Chat | 28.36 | 28.69 | **29.29** |
> | Mistral-7B-Instruct | 36.10 | 35.65 | **37.33** |
> | Gemma | 31.60 | 30.84 | **32.38** |
> | Qwen2.5 | 40.43 | 38.02 | **41.68** |
>
> Compared with uniform allocation, pyramid allocation only shows modest improvements on the Llama2 model, it actually suffers performance degradation on other models. This demonstrates a key limitation of fixed-pattern allocation strategies: they rely on prior observations that may not generalize across different model architectures due to varying attention patterns. In contrast, our P2A strategy consistently outperforms both uniform and pyramid allocation across all tested models. This consistent improvement stems from P2A's ability to effectively measure layer-specific cache preferences by analyzing both spatial and temporal characteristics of attention patterns, enabling it to adaptively allocate appropriate cache sizes to corresponding layers.
>
> >#### **W3**:  Compatibility with Flash-Attention.
>
> We appreciate the reviewer's concern but want to clarify that CAKE is directly implemented on top of Flash-Attention.
>
> While implementing our KV cache eviction method, we still use Flash-Attention's "`_flash_attention_forward`" for full attention computations. The only additional computation needed is to obtain attention weights from the observing window $\mathbf{A}[-S_w:,:]$ for calculating preference scores and eviction indicators, which can be efficiently computed via $\mathbf{A}[-S_w:,:] = \text{Softmax}(\frac{\mathbf{Q}[-S_w:,:]\mathbf{K}^T}{\sqrt{D}})$. For a 32K input sequence, this local attention computation introduces negligible overhead (0.1% of full attention). Therefore, CAKE fully preserves the efficiency benefits of Flash-Attention while providing additional memory optimization through adaptive cache management. To better illustrate the compatibility, we provide a detailed PyTorch-style implementation with Flash-Attention in **Appendix C**, Listing 1.

---

> > ### Comment · Reviewer_zgXb · 2024-11-27
> > **Thanks for your responses**
> >
> > I would like to thank the authors for the detailed responses to my questions, especially for conducting large amounts of additional experiments, which is very impressive. I will raise my score accordingly.

---

> > > ### Author Response · Authors · 2024-11-27
> > > **Response to reviewer zgXb**
> > >
> > > We're delighted that our additional experiments and explanations have sufficiently addressed your concerns.  Once again, we would like to express our sincere gratitude for your positive feedback and for contributing to our manuscript.

---

> ### Author Response · Authors · 2024-11-23
> **Response to reviewer zgXb, part2**
>
> >#### **W4**: Evaluation on more backbones.
>
> Following your advice, we have incorporated experiments on two additional backbones, Qwen and Gemma, to further strengthen our analysis. Specifically, we conducted comparisons under both low-budget settings ($B_{\text{total}} = 128L$) and high-budget settings $(B_{\text{total}} = 1024L)$ across 16 datasets on LongBench. The average results are presented as follows, with detailed results provided in **Appendix F.2**.
>
> Method | Qwen2.5-7B-Instruct | Gemma-7B-Instruct
> -------|---------------------|-------------------
> Full Cache | 48.87 | 34.09
> **Cache size = 128L** | |
> StreamingLLM | 31.32 | 25.68
> H2O | 38.3 | 30.49
> TOVA | 38.0 | 31.24
> SnapKV | 40.09 | 31.57
> PyramidKV | 36.83 | 30.61
> CAKE (ours) | **41.68** | **32.38**
> **Cache size = 1024L** | |
> StreamingLLM | 37.38 | 29.73
> H2O | 44.14 | 33.32
> TOVA | 46.23 | 34.07
> SnapKV | 47.27 | 34.05
> PyramidKV | 45.85 | 33.44
> CAKE (ours) | **47.70** | **34.18**
>
> As can be seen, CAKE consistently outperforms other baselines across both low-memory and high-memory scenarios, even with the inclusion of the new backbones. Notably, Gemma with $B_{\text{total}} = 1024L$ achieves a performance that surpasses the full-cache baseline (34.18 vs. 34.09). In addition to expanding the range of model architectures, we also conducted experiments with larger model sizes, including 13B, 32B, and 70B. CAKE continues to deliver the best performance in these settings, with detailed experimental results available in **Appendix F.2**. We believe these results highlight the effectiveness and generalizability of our proposed approach.
>
> >#### **W5**: Code Reproducibility
>
> We appreciate your valuable feedback regarding code availability and fully understand the importance of open-sourcing for ensuring reproducibility. To address this, we are actively preparing the code and relevant documentation for public release. We will ensure that our work can be fully reproduced by the research community and plan to make the codebase available upon the acceptance of this paper.
>
> >#### **W6**: Comparison with quantization methods.
>
> We appreciate the reviewer's suggestion. It's important to note that CAKE focuses on KV cache eviction through dropping unimportant KV pairs, which is orthogonal to KV cache quantization methods that aim to reduce storage overhead through bit reduction. Both CAKE and quantization methods can be jointly used with flash-attention. More importantly, CAKE is compatible with quantization methods to pursue more efficient KV cache storage as we evaluate in the following part. We have conducted additional experiments on Llama2-7B-Chat comparing CAKE with two typical quantization methods: 1) KIVI[1], a state-of-the-art KV cache quantization method that adopts asymmetric KCVT quantization (quantizes Key cache per-channel and Value cache per-token, similar patterns are also adopted by KVQuant [2] and GEAR [3]), and 2) KCVC, which quantizes KV cache both per-channel for efficiency. Due to time constraints during rebuttal, we compare with KIVI as a representative method, since it shares similar quantization schemes with KVQuant and GEAR. The experimental results on LongBench are summarized as follows:
>
> Method | Compression ratio | Avg.
> -------|------------------|------
> Full Cache (16 bit) | - | 33.07
> **CAKE** | 50% | **33.23**
> KCVC (4 bit) | 25% | 32.72
> KIVI (4 bit) | 25% | 32.71
> **CAKE** | 25% | **32.91**
> KCVC (2 bit) | 12.5% | 23.71
> KIVI (2 bit) | 12.5% | 32.17
> **CAKE** | 12.5% | **32.32**
> KIVI (4 bit) + **CAKE** | 12.5% | **32.51**
> KIVI (4 bit) + **CAKE** | 6.75% | **32.48**
>
> For a fair comparison, we evaluate different methods under the same compression ratios. CAKE achieves better performance (33.23) than full cache (33.07), at a compression ratio of 50%; At 25% and 12.5% compression ratios, CAKE consistently outperforms both KIVI and KCVC; Most importantly, combining CAKE with KIVI-INT4 achieves better results at 12.5% and 6.25% compression ratios compared to KIVI-INT2 alone. These results validate that: **(a)** CAKE is effective as a standalone method, **(b)** CAKE can work synergistically with quantization approaches, **(c)** The combination enables even higher compression ratios while maintaining performance.  We have added this discussion in **Appendix E**.
>
> [1] Liu, Zirui, et al., "KIVI: Plug-and-Play 2bit KV Cache Quantization with Streaming Asymmetric Quantization." (2024).
>
> [2] Hooper, Coleman, et al., "KVQuant: Towards 10 Million Context Length LLM Inference with KV Cache Quantization." arXiv preprint arXiv:2401.18079 (2024).
>
> [3] GEAR: An Efficient KV Cache Compression Recipe for Near-Lossless Generative Inference of LLM.

---

> > ### Comment · Area_Chair_Lr3e · 2024-11-26
> >
> > Dear reviewer zgXb,
> >
> > Could you please respond to authors' rebuttal and see if you would like to update your review? Thanks very much!
> >
> > AC

---

### Official Review · Reviewer_BrWg · 2024-11-02

**Soundness:** 4
**Presentation:** 4
**Contribution:** 4
**Rating:** 10
**Confidence:** 5

**Summary:**

A novel technique that solves KV cache management to improve computational efficiency considering spatial and temporal attention dynamics. 10× faster decoding for extended sequences. Layer-wise memory budget allocation.

**Strengths:**

S1. Identified and visualized the attention dynamics in spatial and temporal axis

S2. Strong empirical analysis on multiple datasets

S3. Actual improvement in inference latency

S4. Excellent presentation (great paper flow and beautiful figures)

**Weaknesses:**

I do not identify any major weaknesses of the paper.

**Questions:**

Q1. I'm just curious how this technique could scale to larger models (no need to verify it empirically)

Q2. Could this technique potentially help transformers stay competitive against RNN-based models like Mamba?

---

> ### Author Response · Authors · 2024-11-23
> **Response to reviewer BrWg**
>
> We are deeply grateful for your thorough review and strong endorsement of our work's technical novelty, presentation quality, and empirical analysis. We appreciate your insightful questions, which we have carefully addressed below.
>
> > **Q1:** How this technique could scale to larger models.
>
> This question is of such high quality that it motivates us to delve into the potential of larger models, even though it was not a requirement from the reviewer.
>
> Although attention patterns vary significantly across different model architectures and sizes, our method can universally analyze their spatial and temporal attention characteristics and adaptively accommodate different model scales and architectures. This adaptability enables our approach to demonstrate robust performance across various model sizes and architectures. To empirically validate this generalizability, we have conducted additional experiments on larger models including Llama2-13B-Chat, Qwen2.5-32B-Instruct, and Llama3-70B-Instruct. We evaluated both low-budget (*128L*) and high-budget ($1024L$) settings on Longbench. The table below presents the average scores across 16 datasets (detailed results are provided in **Appendix F.3**).
>
> | Method | Llama2-13B-Chat | Qwen2.5-32B-Instruct | Llama3-70B-Instruct |
> |:---:|:---:|:---:|:---:|
> | Full Cache | 29.95 | 48.39 | 45.79 |
> |**Cache budget = 128L**|
> | StreamingLLM | 22.75 | 33.33 | 37.53 |
> | H2O | 25.15 | 38.96 | 40.13 |
> | TOVA | 25.01 | 40.08 | 34.76 |
> | SnapKV | 25.85 | 40.56 | 41.20 |
> | PyramidKV | 25.95 | 38.51 | 40.88 |
> | CAKE (ours) | **26.56** | **41.30** | **42.62** |
> |**Cache budget = 1024L**|
> | StreamingLLM | 27.32 | 38.74 | 41.86 |
> | H2O | 28.76 | 43.89 | 44.25 |
> | TOVA | 29.10 | 46.47 | 45.03 |
> | SnapKV | 29.60 | 47.17 | 45.20 |
> | PyramidKV | 29.84 | 46.73 | 45.08 |
> | CAKE (ours) | **29.98** | **47.59** | **45.83** |
>
> Across different model sizes, CAKE consistently outperforms other methods, and notably, under the 1024L setting, CAKE even achieves better performance than full-cache settings for both Llama2-13B and Llama3-70B. We have further validated our method's effectiveness on **additional model architectures**, including Qwen and Gemma, with detailed results available in **Appendix F.2**.
>
> > **Q2:** Could this technique potentially help transformers stay competitive against RNN-based models like Mamba?
>
> Yes, CAKE could potentially help transformers remain competitive against RNN-based models like Mamba by addressing several critical challenges:
>
> 1.  **Memory Efficiency:** While Mamba achieves linear memory scaling, CAKE significantly reduces the Transformer's memory footprint by maintaining a fixed KV cache budget without sacrificing performance. Additionally, CAKE is compatible with efficient attention mechanisms such as FlashAttention, further improving inference efficiency.
>
> 2.   **Decoding Speed:** CAKE enables faster decoding for long sequences (up to 10x speedup for 128K sequences) through optimized cache management, narrowing the speed gap with Mamba.
>
> 3. **Attention Capabilities:** A key advantage of transformer-based models over Mamba lies in their stronger ability to handle complex contexts. CAKE preserves this strength while enhancing efficiency.
>
> While CAKE makes LLMs more efficient for generation, our analysis reveals that attention mechanisms can exhibit redundancy, as only a subset of information is required for effective inference. This insight suggests the potential for combining the strengths of both architectures to develop a more efficient hybrid framework.

---

> > ### Comment · Reviewer_BrWg · 2024-11-26
> >
> > Thank you for the responses. I believe this paper deserves more recognition.

---

> > > ### Author Response · Authors · 2024-11-26
> > > **Response to reviewer BrWg**
> > >
> > > Thank you for your encouraging feedback and support of our research. We sincerely appreciate your recognition and the time you dedicated to reviewing our work!

---

### Official Review · Reviewer_LJXW · 2024-11-02

**Soundness:** 3
**Presentation:** 3
**Contribution:** 2
**Rating:** 6
**Confidence:** 2

**Summary:**

This paper introduces a method for optimizing KV cache eviction through a cache allocation strategy to enhance LLM inference efficiency. The proposed cache allocation adapts to layer preferences, adjusting KV cache injection to improve efficiency while maintaining satisfactory performance. Extensive experiments are conducted to demonstrate the method’s effectiveness.

**Strengths:**

The paper is well-written, with clear motivations for the proposed method and pipeline illustrations. Incorporating layer-wise preference modeling to guide KV caching strategies is intuitive, given the insights from attention dynamics analysis.

The proposed method is straightforward and compatible with existing KV caching strategies, making it easy to integrate while achieving decent efficiency. The authors provide ample experiments to substantiate this point.

Empirical analysis is comprehensive, covering 16 tasks with various LLMs of different specifications, offering a comprehensive evaluation.

**Weaknesses:**

Most empirical analyses focus on smaller LLMs with 7B-8B parameters, which may limit the generalizability of this approach for much larger LLMs. Specifically, it would be valuable to see how computational costs and performance are impacted across different LLM sizes.

The empirical improvements over existing baselines are relatively modest, which could suggest limited practical advantages in some cases.

**Questions:**

Please see weakness aspects above.

---

> ### Author Response · Authors · 2024-11-23
> **Response to reviewer LJXW**
>
> We sincerely appreciate your positive assessment of our work. Your valuable feedback has helped us improve the quality and completeness of our paper. We have carefully addressed each of your comments below, with the corresponding modifications **highlighted in blue** in the revised manuscript.
>
> > **W1:** Most empirical analyses focus on smaller LLMs with 7B-8B parameters, which may limit the generalizability of this approach for much larger LLMs.
>
> To demonstrate the generalizability of our approach across different model sizes, we have conducted additional experiments on larger models including Llama2-13B-Chat, Qwen2.5-32B-Instruct, and Llama3-70B-Instruct. We evaluated both low-budget ( cache size = $128L$) and high-budget (cache size = $1024L$) settings on Longbench. The table below presents the average scores across 16 datasets (detailed results are provided in **Appendix F.3**).
> | Method | Llama2-13B-Chat | Qwen2.5-32B-Instruct | Llama3-70B-Instruct |
> |:---:|:---:|:---:|:---:|
> | Full Cache | 29.95 | 48.39 | 45.79 |
> | **Cache size = 128L**|
> | StreamingLLM | 22.75 | 33.33 | 37.53 |
> | H2O | 25.15 | 38.96 | 40.13 |
> | TOVA | 25.01 | 40.08 | 34.76 |
> | SnapKV | 25.85 | 40.56 | 41.20 |
> | PyramidKV | 25.95 | 38.51 | 40.88 |
> | CAKE (ours) | **26.56** | **41.30** | **42.62** |
> |**Cache size = 1024L** |
> | StreamingLLM | 27.32 | 38.74 | 41.86 |
> | H2O | 28.76 | 43.89 | 44.25 |
> | TOVA | 29.10 | 46.47 | 45.03 |
> | SnapKV | 29.60 | 47.17 | 45.20 |
> | PyramidKV | 29.84 | 46.73 | 45.08 |
> | CAKE (ours) | **29.98** | **47.59** | **45.83** |
>
> As shown in the results, CAKE consistently outperforms baseline methods across different model sizes. Under constrained memory conditions (cache size = $128L$), CAKE demonstrates significant advantages over other methods. These advantages are maintained with larger cache sizes ($1024L$), where CAKE even achieves slightly better performance than full-cache settings for some models (Llama2-13B: 29.98 vs 29.95, and Llama3-70B: 45.83 vs 45.79). These results indicate that our approach scales effectively to larger models while maintaining its efficiency advantages.
>
> Additionally, we have validated CAKE's effectiveness on **two more model architectures**, including Qwen and Gemma, further demonstrating the generalizability of our proposed method. Detailed results can be found in **Appendix F.2**.
>
> > **W2:** The empirical improvements over existing baselines are relatively modest, which could suggest limited practical advantages in some cases.
>
> Our initial submission might show modest improvements in some aspects. However, by incorporating our new experiments to address **W1**, we wish to emphasize that CAKE offers significant practical benefits in four critical areas:
>
> 1. CAKE consistently outperforms baselines across different model architectures (Llama, Mistral, Qwen, Gemma) ranging from 7B to 70B parameters on LongBench across all memory settings (**Appendix F**).
>
> 2. CAKE shows significant advantages in memory-constrained settings. For example, with cache size $64L$ on Mistral, CAKE achieves 34.31 average score on LongBench versus SnapKV's 31.31 and PyramidKV's 30.50 (Table 7, **lines 1147-1149**), highlighting its efficiency in resource-limited scenarios. Similar cases can be also found in other models.
>
> 3. CAKE not only narrows the gap with full cache but sometimes surpasses it while maintaining minimal memory, as demonstrated on Gemma-7B, Llama2-13B, and Llama3-70B with only $1024L$ cache size.
>
> 4. CAKE significantly outperforms existing methods on challenging Multi-Retrieval tasks. For example, with $1024$ cache size on Mistral, CAKE achieves 71.00 versus SnapKV's 56.93 (Table 10, lines **1340-1342**) and maintains 48.55 accuracy versus SnapKV's 19.14 (Table 10, **lines 1344-1346**) with $512$ cache size. More similar cases can be found in **Appendix G**.

---

> > ### Comment · Area_Chair_Lr3e · 2024-11-26
> >
> > Dear reviewer LJXW,
> >
> > Could you please respond to authors' rebuttal and see if you would like to update your review? Thanks very much!
> >
> > AC

---

> > ### Comment · Reviewer_LJXW · 2024-11-26
> > **Thanks for the rebuttal.**
> >
> > Thank you for including additional experiments on larger LLMs and I will keep my positive evaluation of your work. I encourage the authors to update their manuscript by incorporating highlights of their empirical improvements into the main body of the paper based on their response to the second question.

---

> > > ### Author Response · Authors · 2024-11-27
> > > **Response to reviewer LJXW**
> > >
> > > Thank you for your valuable feedback! We have revised the manuscript to better emphasize our key contributions.  We're delighted that our additional experiments and explanations have sufficiently addressed your concerns.  Once again, we would like to express our sincere gratitude for your positive feedback and for contributing to our manuscript.

---

### Official Review · Reviewer_rCQd · 2024-11-11

**Soundness:** 3
**Presentation:** 2
**Contribution:** 3
**Rating:** 6
**Confidence:** 3

**Summary:**

This paper proposes Cascading and Adaptive Key-value cache Eviction (CAKE) method for optimizing Key-Value cache evicting in large language models. Specifically, CAKE assesses each layer’s KV cache needs by considering attention dynamics in both spatial and temporal dimensions. During the prompt prefilling, CAKE allocates rational cache size for layers by analyzing layer-specific KV cache preferences and manages the memory budgets with the guidance of these preferences in a cascading manner. Besides, CAKE introduces a novel eviction indicator that accounts for both the long-term influence and temporal variability of token importance. Extensive experiments demonstrate CAKE’s superior performance across different models and memory constraints, especially in low-memory scenarios.

**Strengths:**

(1)	This paper provides a novel and practical key-value cache eviction approach to enhance LLM’s proficiency in handling long sequences, based on layer-specific KV cache preferences in a cascading manner.

(2)	The paper is well-organized, and the writing is clear. In particular, Figure 2 clearly points out the differences between the proposed CAKE and other existing models, and I find almost no typos in this paper.

(3)	The theoretical analysis (e.g. Theorem 2) rigorously demonstrates the equivalent KV cache eviction results of the proposed preference-guided cascading cache management to the vanilla preference-prioritized adaptive allocation strategy.

(4)	The extensive experiments on several open-source LLM benchmarks truly validate the effectiveness of the proposed algorithms.

(5)	Overall, the proposed algorithm CAKE model is novel, practical and efficient. The corresponding experimental results are extensive and sound.

**Weaknesses:**

(1)	The abbreviation “KV” in the title and abstract (e.g. line 12, or the second line of abstract) should be clearly written as “Key-value”, as this word appears for the first time in the whole paper.

(2)	In line 23, the abstract section says “this approach allows for a global view of cache size allocation, distributing resources OPTIMALLY”. The word “optimally” is somewhat controversial, until you can demonstrate rigorously from a theoretical point that the proposed resources distribution method is optimal (with respect to certain theoretical property). Therefore, I would suggest to use less controversial word, like “adaptively”.

(3)	Equation (4) is a little confusing for me. Since $A[i,:]$ is a row vector, $\log{A[i,:]}$ is also a row vector, then is $ A[i,:] \log{A[i,:]}$ the inner product of these two vectors? Should give more clear explanations.

(4)	I read the proofs line by line, and according to my experience, the proof is sound. However, since the proof of Theorem 1 is truly basic and short, I believe it will be better to describe “Theorem 1” as a proposition. Besides, in Theorem 1, “For layer $l \in [L]$”, does it mean for any (fixed) layer, or mean there exists a layer? More explanations should be clarified.

(5)	In Table 1 of the experimental results, the proposed CAKE obtains SOTA results in most benchmark datasets. But in some benchmarks, some existing models like TOVA and SnapKV can achieve better results. Could you give more analysis in which kind of benchmark datasets can CAKE obtain better experimental results than the existing baselines?

**Questions:**

In Table 1, CAKE could not outperform existing methods on some benchmarks. Could you give more analysis in which kind of benchmark datasets can CAKE obtain better experimental results than the existing baselines? Or in what situations, CAKE will fail to achieve good results?

---

> ### Author Response · Authors · 2024-11-23
> **Response to reviewer rCQd**
>
> We sincerely appreciate your thorough review and positive assessment of our work. Your constructive feedback is invaluable in helping us improve both the technical content and presentation clarity of our paper. We have carefully addressed each of your comments below, with the corresponding modifications **highlighted in blue** in the revised manuscript.
>
> > **W1&W2**: Suggestions on Terminology and Wording
>
>  We appreciate your valuable feedback on our writing. We agree with both suggestions and will revise "KV" to "Key-value (KV)" on its first appearance in the abstract. We also concur that replacing "optimally" with "adaptively" more accurately reflects our method's capabilities. Thank you for helping us improve the precision and clarity of our paper. These changes have been incorporated in the revised version of our paper.
>
> > **W3**: Clarification on Equation (4).
>
>  We regret the error in our previous statement. The correct operation involves transposing $\text{log}\mathbf{A}[i, :]$ and then computing the inner product with $\mathbf{A}[i, :]$ by using the expression $\mathbf{A}[i, :]\text{log}(\mathbf{A}[i,:])^T$. We have made the necessary correction in our paper and are grateful for your meticulous review.
>
> > **W4**:  Clarification on Theorem 1.
>
> Thank you for your careful review of the mathematical presentation. We have changed "Theorem 1" to "Proposition 1" given its straightforward proof. In Proposition 1, "For layer $l∈[L]$" means the allocated budget size decreases monotonically from stage $l$ to $L-1$ for any layer with index in $[0,1,...,L-1]$. We have modified this to "For any layer $l∈[L]$" for better clarity.
>
> > **W5 & Q1**: Could you give more analysis in which kind of benchmark datasets can CAKE obtain better experimental results than the existing baselines?
>
> We appreciate your insightful question. This is an excellent point that helps us better articulate the comprehensive strengths of CAKE across diverse benchmarks. Our analysis demonstrates CAKE's robust performance and specific advantages in different types of evaluations:
>
> On LongBench tasks (detailed in Table 1 and **Appendix F**), CAKE demonstrates consistent performance improvements across various cache sizes and task types. While other methods occasionally show marginal advantages in specific cases, none exhibits consistent superiority across all conditions, highlighting CAKE's robust general performance.
>
> For NeedleBench (detailed in **Appendix G**), CAKE's advantages become more pronounced in complex tasks, particularly in Multi-Needle Retrieval. For instance, compared to the previous SOTA SnapKV, CAKE achieves significantly better results on Mistral: 71.00 vs. 56.93 ($B_\text{total}=1024L$, **lines 1340-1342**) and 48.55 vs. 19.14 ($B_\text{total}=512L$，**lines 1344-1346**) as shown in **Table 10** of Appendix G. Such advantage becomes increasingly pronounced as the cache budget decreases. CAKE's superior performance in these challenging scenarios stems from its design, which considers both long-term significance and short-term fluctuations in information relevance. In contrast to existing methods, CAKE employs a more nuanced approach, maintaining a more comprehensive and balanced representation of contextual information, instead of depending solely on static attention score or fixed cache allocation strategies, thus avoiding premature discarding of information that could be vital for subsequent complex retrievals.
>
> Nevertheless, as an eviction strategy, under extremely constrained budgets, CAKE's performance inevitably experiences some degradation. To address this, we suggest combining KV Cache quantization with CAKE eviction. Our experiments illustrate that this hybrid approach proves adept at maintaining performance under severe memory limitations, as detailed in **Appendix E**.

---

> ### Comment · Area_Chair_Lr3e · 2024-11-26
>
> Dear reviewer rCQd,
>
> Could you please respond to authors' rebuttal and see if you would like to update your review? Thanks very much!
>
> AC

---

### Author Response · Authors · 2024-11-23
**Summary and general reply to the reviewers**

We sincerely appreciate all reviewers' time and efforts in reviewing our paper. Your constructive feedback has substantially helped improve the quality of our work.
We are particularly encouraged that the reviewers have recognized our key contributions in several aspects:

* Novel and Practical Contribution (Reviewer-rCQd, BrWg, zgXb)

* Strong Technical Foundation (Reviewer-rCQd, LJXW, BrWg)
* Comprehensive Empirical Validation (Reviewer-rCQd, LJXW, BrWg)
* Clear Presentation and Organization (Reviewer-rCQd, LJXW, BrWg, zgXb)

In response to the reviewers' suggestions, we have made the following major improvements:

* Extended evaluation to **additional LLM architectures** (Qwen and Gemma), with detailed results provided in **Appendix F.2**. (Reviewer-zgXb)
* Conducted experiments on **larger models** ranging from 13B to 70B parameters (Llama2-13B, Qwen2.5-32B, and Llama3-70B), with comprehensive results presented in **Appendix F.3**. (Reviewer-LJXW, BrWg)
* Added discussion on orthogonal KV cache quantization methods in **Appendix E**, demonstrating that our work is **compatible with and complementary** to these approaches. (Reviewer-zgXb)
* Addressed all other clarification requests from the reviewers. (Reviewer-rCQd, zgXb)

All major modifications have been **highlighted in blue** in the revised manuscript for easy reference. We believe these changes have significantly strengthened our paper and addressed the reviewers' concerns.

---

### Author Response · Authors · 2024-11-25
**Kind Request for Discussions and Feedback for Paper 9488**

Dear Reviewers,

We are deeply grateful for your thorough reviews and valuable feedback on our paper.

As the discussion period nears its end, we hope our responses have effectively addressed the points you raised. Your insights have been instrumental in strengthening our work, and we welcome your continued engagement.

Thank you for your dedication and expertise.

Best regards,

Authors of Paper 9488

---

### Meta-Review · Area_Chair_Lr3e · 2024-12-23

**Metareview:**

All reviewers agreed the paper proposed a useful contribution to KV cache eviction strategy to speed up decoding process of LLM.
Strength:
1. The proposal method appeared to be novel and useful.
2. Good experimental results

Weakness:
1. Results were only on small models, limiting its impact on practical use with larger models (more results came in rebuttal period).
2. Baseline for comparisons were not particulartly strong. (Improvements were made during rebuttal.)

**Additional Comments On Reviewer Discussion:**

Most reviewers actively participated in the discussion and concerns were mostly addressed by the rebuttal. (I discounted rating 10 as the comments were mostly subjective in the strength section.)

---

### Decision · Program_Chairs · 2025-01-22

Accept (Poster)